# Bayesian Risk Markov Decision Processes

**Yifan Lin**
Industrial and Systems Engineering
Georgia Institute of Technology
Atlanta, GA 30332, USA
ylin429@gatech.edu

**Yuxuan Ren**
Industrial and Systems Engineering
Georgia Institute of Technology
Atlanta, GA 30332, USA
yren79@gatech.edu

**Enlu Zhou**
Industrial and Systems Engineering
Georgia Institute of Technology
Atlanta, GA 30332, USA
enlu.zhou@isye.gatech.edu

## Abstract

We consider finite-horizon Markov Decision Processes where parameters, such as transition probabilities, are unknown and estimated from data. The popular distributionally robust approach to addressing the parameter uncertainty can sometimes be overly conservative. In this paper, we propose a new formulation, Bayesian risk Markov decision process (BR-MDP), to address parameter uncertainty in MDPs, where a risk functional is applied in nested form to the expected total cost with respect to the Bayesian posterior distributions of the unknown parameters. The proposed formulation provides more flexible risk attitudes towards parameter uncertainty and takes into account the availability of data in future time stages. To solve the proposed formulation with the conditional value-at-risk (CVaR) risk functional, we propose an efficient approximation algorithm by deriving an analytical approximation of the value function and utilizing the convexity of CVaR. We demonstrate the empirical performance of the BR-MDP formulation and proposed algorithms on a gambler's betting problem and an inventory control problem.

## 1 Introduction

Markov decision process (MDP) is a paradigm for modeling sequential decision making under uncertainty. From a modeling perspective, some parameters of MDPs are unknown and need to be estimated from data. In this paper, we consider MDPs where transition probability and cost parameters are not known. A natural question would be: given a finite and probably small set of data, how does a decision maker find an "optimal" policy that minimizes the expected total cost under the uncertain transition probability and cost parameters?

A possible approach that mitigates the parameter uncertainty (also known as epistemic uncertainty) lies in the framework of distributionally robust MDPs (DR-MDPs, [1]). DR-MDP regards the unknown parameters as random variables and assumes the associated distributions belong to an ambiguity set that is constructed from the data. DR-MDP then finds the optimal policy that minimizes the expected total cost with the parameters following the most adversarial distribution within the ambiguity set. However, distributionally robust approaches might yield overly conservative solutions that perform poorly for scenarios that are more likely to happen than the worst case. Moreover, as pointed out in [2], DR-MDP does not explicitly specify the dynamics of the considered problem in the sense that the distribution of the unknown parameters does not depend on realizations of the data process, and therefore is generally not time consistent (we refer the reader to [2] for details on time consistency).

36th Conference on Neural Information Processing Systems (NeurIPS 2022).

In view of the aforementioned drawbacks of DR-MDP, we propose a new formulation named as Bayesian risk MDP (BR-MDP), to address the parameter uncertainty in MDPs. BR-MDP takes a similar perspective as Bayesian risk optimization (BRO), which is a new framework proposed by [3, 4] for static (single stage) optimization. They quantify the parameter uncertainty using a Bayesian posterior distribution to replace the ambiguity set in DRO, and impose a risk functional on the objective function with respect to the posterior distribution. To extend to BRO to MDPs and also ensure time consistency, we propose to use a nested risk functional taken with respect to the process of Bayesian posterior distributions, where the posterior is updated with all realizations of randomness up to each time stage. We show the proposed BR-MDP formulation is time consistent, and derive the corresponding dynamic programming equation with an augmented state that incorporates the posterior information. The proposed framework works for an offline planning problem, where the decision maker, after loaded with the learned optimal policy and put to the real environment, acts like it adapts to the environment.

To solve the proposed BR-MDP formulation, we develop an efficient algorithm by drawing a connection between BR-MDP and partially observable MDP (POMDP) and utilizing the convexity of CVaR. The connection between BR-MDP and POMDP is motivated by the observation that the posterior distribution in BR-MDP is exactly like the belief state (which is the posterior distribution of the unobserved state given the history of observations) in a POMDP. The optimal value function of a POMDP (in a minimization problem) can be expressed as an lower envelope of a set of linear functions (also called $\alpha$-functions, see [5]). We show a similar $\alpha$-function representation of the value function for BR-MDP, where the number of $\alpha$-functions grow exponentially over time. To have computationally feasible algorithm, we further derive an analytical approximation of the value function that keeps a constant number of $\alpha$-functions over time.

To summarize, the contributions of this paper are two folds. First, we propose a new time-consistent formulation BR-MDP to handle the parameter uncertainty in MDPs. Second, we propose an efficient algorithm to solve the proposed formulation with a CVaR risk functional, and the algorithm can be easily extended to other coherent risk measures (see [6] for an overview on coherent risk measures).

## 2 Related Literature

One possible approach that mitigates the parameter uncertainty in MDPs lies in the framework of robust MDPs (e.g. [7, 8, 9, 10, 11]). In robust MDPs, parameters are usually assumed to belong to a known set referred to as the ambiguity set, and the optimal decisions are chosen according to their performance under the worst possible parameter realizations within the ambiguity set. [1] further extends the distributionally robust approach to MDPs (DR-MDPs) with parameter uncertainty and utilizes the probabilistic information of the unknown parameters. Different from all the aforementioned works, we take a Bayesian perspective in the BR-MDP formulation and seek a trade-off between the posterior expected performance and the robustness in the actual performance.

It is worth noting that applying the Bayesian approach to MDPs has been considered in [12], which proposes a Bayes-adaptive MDP (BAMDP) formulation with an augmented state composed of the underlying MDP state and the posterior distribution of the unknown parameters. In BAMDP, each transition probability is treated as an unknown parameter associated with a Dirichlet prior distribution, and an expectation is taken with respect to the Dirichlet posterior on the expected total cost. In contrast, our BR-MDP formulation imposes a risk functional, taken with respect to the posterior distribution (which could be chosen as Dirichlet distribution but is more general), on the expected total cost in a nested form. On a related note, risk-averse decision making has been widely studied in MDPs. Apart from the robust MDPs that address the parameter uncertainty, risk-sensitive MDPs (e.g. [13, 14, 15, 16]) address the intrinsic uncertainty (also known as aleatoric uncertainty) that is due to the inherent stochasticity of the underlying MDP, by replacing the risk-neutral expectation (with respect to the state transition) by general risk measures, such as conditional value-at-risk (CVaR, see [17]). Most of the existing literature on risk-sensitive MDPs consider a static risk functional applied to the total return (e.g. [18, 19, 20, 21]). There are two recent works closely related to ours, both of which apply a risk functional to BAMDP. Specifically, [22] formulates the risk-sensitive planning problem as a two-player zero-sum game and then applies a static risk functional to the expected total cost, which adds robustness to the incorrect priors over model parameters; [23] optimizes a CVaR risk functional over the total cost and simultaneously addresses both epistemic and aleatoric uncertainty. In contrast, we consider a nested risk functional due to the time consistency consideration discussed

in previous section. Also note that our problem setting relies on partial knowledge of the model (state transition equation, cost function etc.) and works in an offline planning setting with no interaction with the environment, and hence is different from Bayesian reinforcement learning (e.g. [24, 25]).

## 3 Preliminaries and Problem Formulation

### 3.1 Preliminaries: BRO and CVaR

Bayesian risk optimization (BRO, see [3, 4]) considers a general stochastic optimization problem: $\min_x \mathbb{E}_{\mathbb{P}^c}[h(x, \xi)]$, where $x$ is the decision vector, $\xi$ is a random vector with distribution $\mathbb{P}^c$, $h$ is a deterministic cost function. The true distribution $\mathbb{P}^c$ is rarely known in practice and often needs to be estimated from data. It is very likely that the solution obtained from solving an estimated model performs badly under the true model. To avoid such a scenario, BRO seeks robustness in the actual performance by imposing a risk functional to the expected cost function and solving the following problem: $\min_x \rho_{\mathbb{P}_n} \{ \mathbb{E}_{\mathbb{P}_\theta}[h(x, \xi)] \}$, where $\rho$ is a risk functional, and $\mathbb{P}_n$ is the posterior distribution of $\theta$ after observing $n$ data points. It is assumed that the unknown distribution $\mathbb{P}^c$ belongs to a parametric family $\{\mathbb{P}_\theta | \theta \in \Theta\}$, where $\Theta$ is the parameter space and $\theta^c \in \Theta$ is the unknown true parameter value. Taking a Bayesian perspective, $\theta^c$ is viewed as a realization of a random vector $\theta$.

In particular, conditional value-at-risk (CVaR), a common coherent risk measure (see [6]), is considered for the risk functional. For a random variable $X$ defined on a probability space $(\Omega, \mathcal{F}, \mathbb{P})$, value-at-risk $\text{VaR}^\alpha(X)$ is defined as the $\alpha$-quantile of $X$, i.e., $\text{VaR}^\alpha(X) := \inf\{t : \mathbb{P}(X \leq t) \geq \alpha\}$, where the confidence level $\alpha \in (0, 1)$. Assuming there is no probability atom at $\text{VaR}^\alpha(X)$, CVaR at confidence level $\alpha$ is defined as the mean of the $\alpha$-tail distribution of $X$, i.e., $\text{CVaR}_\alpha(X) = \mathbb{E}[X \mid X \geq \text{VaR}_\alpha(X)]$. It is shown in [17] that CVaR can be written as a convex optimization:

$$\text{CVaR}_\alpha(X) = \min_{u \in \mathbb{R}} \left\{ u + \frac{1}{1 - \alpha} \mathbb{E}\left[(X - u)^+\right] \right\}, \tag{1}$$

where $(\cdot)^+$ stands for $\max(0, \cdot)$.

### 3.2 New formulation: Bayesian risk MDPs (BR-MDPs)

Consider a finite-horizon MDP defined as $(\mathcal{S}, \mathcal{A}, \mathcal{P}, \mathcal{C})$, where $\mathcal{S}$ is the state space, $\mathcal{A}$ is the action space, $\mathcal{P}$ is the transition probability with $\mathcal{P}(s_{t+1}|s_t, a_t)$ denoting the probability of transitioning to state $s_{t+1}$ from state $s_t$ when action $a_t$ is taken, $\mathcal{C}$ is the cost function with $\mathcal{C}_t(s_t, a_t, s_{t+1})$ denoting the cost at time stage $t$ when action $a_t$ is taken and state transitions from $s_t$ to $s_{t+1}$. A Markovian deterministic policy $\pi$ is a function mapping from $\mathcal{S}$ to $\mathcal{A}$. Given an initial state $s_0$, the goal is to find an optimal policy that minimizes the expected total cost: $\min_\pi \mathbb{E}^{\pi, \mathcal{P}, \mathcal{C}} \left[ \sum_{t=0}^{T-1} \mathcal{C}_t(s_t, a_t, s_{t+1}) \right]$, where $\mathbb{E}^{\pi, \mathcal{P}, \mathcal{C}}$ is the expectation with policy $\pi$ when the transition probability is $\mathcal{P}$ and the cost is $\mathcal{C}$. In practice, $\mathcal{P}$ and $\mathcal{C}$ are often unknown and estimated from data.

To deal with the parameter uncertainty in MDPs, we propose a new formulation, Bayesian risk MDP (BR-MDP), by extending BRO in static optimization to MDPs. We assume the state transition is specified by the state equation $s_{t+1} = g_t(s_t, a_t, \xi_t)$ with a known transition function $g_t$, which involves state $s_t \in \mathcal{S} \subseteq \mathbb{R}^s$, action $a_t \in \mathcal{A} \subseteq \mathbb{R}^a$, and randomness $\xi_t \in \Xi \subseteq \mathbb{R}^k$, where $s, a, k$ are the dimensions of the state, action, and randomness, respectively. We assume $\{\xi_t, t = 0, \cdots, T-1\}$ are independent and identically distributed (i.i.d.). Note that the state equation together with the distribution of $\xi_t$ uniquely determines the transition probability of the MDP, i.e., $\mathcal{P}(s_{t+1} \in S'|s_t, a_t) = \mathbb{P}(\{\xi_t \in \Xi : g_t(s_t, a_t, \xi_t) \in S'\}|s_t, a_t)$, where $S'$ is a measurable set in $\mathcal{S}$. We refer the readers to Chapter 3.5 in [26] for the equivalence between stochastic optimal control and MDP formulation. We use the representation of state equations instead of transition probabilities in MDPs, for the purpose of decoupling the randomness and the policy, leading to a cleaner formulation in the nested form. We assume the distribution of $\xi_t$, denoted by $f(\cdot; \theta^c)$, belongs to a parametric family $\{f(\cdot; \theta) | \theta \in \Theta\}$, where $\Theta \subseteq \mathbb{R}^d$ is the parameter space, $d$ is the dimension of the parameter $\theta$, and $\theta^c \in \Theta$ is the true but unknown parameter value. The parametric assumption is satisfied in many problems; for example, in inventory control the customer demand is often assumed to follow a Poisson process (see [27]) with unknown arrival rate. The cost at time stage $t$ is assumed to be a function of state $s_t$, action $a_t$, and randomness $\xi_t$, i.e., $\mathcal{C}_t(s_t, a_t, \xi_t)$.

We start with a prior distribution $\mu_0$ over the parameter space $\Theta$, which captures the initial uncertainty in the parameter estimate from an initial data set and can also incorporate expert opinion. Then given an observed realization of the data process, we update the posterior distribution $\mu_t$ according to the Bayes' rule. Let the policy be a sequence of mappings from state $s_t$ and posterior $\mu_t$ to the action space, i.e., $\pi = \{\pi_t | \pi_t : \mathcal{S} \times \mathcal{M}_t \to \mathcal{A}, t = 0, \cdots, T-1\}$, where $\mathcal{M}_t$ is the space of posterior distributions at time stage $t$. Now we present the BR-MDP formulation below.

$$\min_{\pi} \ \rho_{\mu_0} \mathbb{E}_{f(\cdot;\theta_0)} \left[ \mathcal{C}_0(s_0, a_0, \xi_0) + \cdots + \rho_{\mu_{T-1}} \mathbb{E}_{f(\cdot;\theta_{T-1})} \left[ \mathcal{C}_{T-1}(s_{T-1}, a_{T-1}, \xi_{T-1}) + \mathcal{C}_T(s_T) \right] \right] \quad (2)$$

$$s.t. \ \ s_{t+1} = g_t(s_t, a_t, \xi_t), \ \ t = 0, \cdots, T-1; \quad (3)$$

$$\mu_{t+1}(\theta) = \frac{\mu_t(\theta) f(\xi_t; \theta)}{\int_\Theta \mu_t(\theta) f(\xi_t; \theta) \, d\theta}, \ t = 0, \cdots, T-1, \quad (4)$$

where $a_t = \pi_t(s_t, \mu_t)$, $\theta_t$ is a random vector following distribution $\mu_t$, $\mathbb{E}_{f(\cdot;\theta_t)}$ denotes the expectation with respect to $\xi_t \sim f(\cdot; \theta_t)$ conditional on $\theta_t$, and $\rho_{\mu_t}$ denotes a risk functional with respect to $\theta_t \sim \mu_t$. We assume the last-stage cost only depends on the state, hence denoted by $\mathcal{C}_T(s_T)$. Equation (3) is the transition of the state $s_t$, and without loss of generality we assume the initial state $s_0$ takes a deterministic value. Equation (4) is the updating of the posterior $\mu_t$ given the prior $\mu_0$.

### 3.3 Time consistency and dynamic programming

It is important to note that the BR-MDP formulation (2) takes a nested form of the risk functional. A primary motivation for considering such nested risk functional is the issue of time consistency (see [28, 29, 30, 2]), which means that the optimal policy solved at time stage 0 is still optimal for any remaining time stage $t \geq 1$ with respect to the conditional risk functional, even when the realization of the randomness $\xi_t$ is revealed up to that time stage. In contrast, optimizing a static risk functional can lead to "time-inconsistent" behavior, where the optimal policy at the current time stage can become suboptimal in the next time stage simply because a new piece of information is revealed (see [28, 29]).

To illustrate, consider the simple case of a three-stage problem where the risk functional $\rho$ is CVaR with confidence level $\alpha$. Our BR-MDP solves a nested formulation

$$\min_{a_0, a_1} \rho_{\mu_0} \mathbb{E}_{f(\cdot;\theta_0)} [\mathcal{C}_0(s_0, a_0, \xi_0) + \rho_{\mu_1} \mathbb{E}_{f(\cdot;\theta_1)} [\mathcal{C}_1(s_1, a_1, \xi_1) + \mathcal{C}_2(s_2)]]$$

while the non-nested counterpart solves

$$\min_{a_0, a_1} \rho_{\mu_0} \mathbb{E} [\mathcal{C}_0(s_0, a_0, \xi_0) + \mathcal{C}_1(s_1, a_1, \xi_1) + \mathcal{C}_2(s_2)],$$

where the expectation is taken with respect to (w.r.t.) the joint distribution of $\xi_0$ and $\xi_1$, and the static risk functional is applied to the total cost. Then we have the following relation between the two formulations:

$$\rho_{\mu_0} \mathbb{E}_{f(\cdot;\theta_0)} [\mathcal{C}_0(s_0, a_0, \xi_0) + \rho_{\mu_1} \mathbb{E}_{f(\cdot;\theta_1)} [\mathcal{C}_1(s_1, a_1, \xi_1) + \mathcal{C}_2(s_2)]]$$
$$\geq \rho_{\mu_0} \mathbb{E}_{f(\cdot;\theta_0)} [\mathcal{C}_0(s_0, a_0, \xi_0) + \mathbb{E}_{\xi_1|\xi_0} \mathcal{C}_1(s_1, a_1, \xi_1) + \mathcal{C}_2(s_2)]$$
$$= \rho_{\mu_0} \mathbb{E} [\mathcal{C}_0(s_0, a_0, \xi_0) + \mathcal{C}_1(s_1, a_1, \xi_1) + \mathcal{C}_2(s_2)],$$

where the inequality is justified by CVaR being the right tail average of the distribution, and the equality follows from the tower property of conditional expectation. The upper bound is used to show that the static risk functional always yields a higher total expect cost than the nested risk functional, illustrating the benefit of nested risk functional which originates from time consistency.

Another drawback of the static formulation for a general risk functional is the lack of dynamic programming equation. In particular, [2] points out that for the static formulation, the derivation of the dynamic programming equation is based on the interchangeability principle and the decomposability property of the risk functional, where such decomposability property holds only for expectation and max-type risk functionals. On the other hand, for our nested formulation (2), the corresponding dynamic programming equation is easily obtained as follows:

$$V_t^*(s_t, \mu_t) = \min_{a_t \in \mathcal{A}} \ \rho_{\mu_t} \mathbb{E}_{f(\cdot;\theta_t)} \left[ \mathcal{C}_t(s_t, a_t, \xi_t) + V_{t+1}^*(s_{t+1}, \mu_{t+1}) | s_t, \mu_t, a_t \right], \forall s_t, \mu_t, \quad (5)$$

where $s_t$ and $\mu_t$ follow equation (3) and (4), respectively. Therefore, our BR-MDP formulation provides a time-consistent risk-averse framework to deal with epistemic uncertainty in MDPs. The exact dynamic programming is summarized in Algorithm 1, for benchmark purpose.

**Algorithm 1:** Exact dynamic programming for finite-horizon BR-MDPs.

---

**input**: finite horizon $T$, initial state $s_0$, prior distribution $\mu_0$;
**output**: optimal value function $V_0^*(s_0, \mu_0)$ and corresponding optimal policy $\pi^*$;
set $V_T^*(s_T, \mu_T) = \mathcal{C}_T(s_T), \forall (s_T, \mu_T) \in \mathcal{S} \times \mathcal{M}_t$;
**for** $t \leftarrow T - 1$ **to** *0* **do**
    **for** *each* $(s_t, \mu_t) \in \mathcal{S} \times \mathcal{M}_t$ **do**
        solve dynamic programming equation (5);
        set $\pi_t^*(s_t, \mu_t) := a_t^*$, where $a_t^*$ attains $V_t^*(s_t, \mu_t)$;
    **end**
**end**

---

## 4 An Analytical Approximate Solution to BR-MDPs

Although the exact dynamic programming works for a general risk functional, there are two challenges to carry it out. First, the expectation and the risk functional are generally impossible to compute analytically and estimation by (nested) Monte Carlo simulation can be computationally expensive. Second, the update of the posterior distribution $\mu_t$ does not have a closed form in general and often results in an infinite-dimensional posterior. We circumvent the latter difficulty by using conjugate families of distributions (see Chapter 5 in [31]), where the posterior distribution falls into the same parametric family as the prior distribution, and thus maintain the dimensionality of the posterior to be the finite (and often small) dimension of the parameter space of the conjugate distribution. However, the posterior parameters usually take continuous values and hence BR-MDP with the augmented state $(s, \mu)$ is a continuous-state MDP. We note that this continuous-state MDP resembles a belief-MDP, which is the equivalent transformation of a POMDP by regarding the posterior distribution of the unobserved state as a (belief) state (see [32]), and our posterior distribution $\mu_t$ is just like the belief state in POMDP in the sense that both are posterior distributions updated via Baye's rule. This observation motivates our algorithm development in this section.

Specifically, we derive an efficient approximation algorithm for the BR-MDP with the risk functional CVaR. The main idea is that for CVaR, once we know the variable $u$ in (1), it is reduced to an expectation of a convex function. If there is a way to approximate the value function for a given $u$, we can utilize the convexity of CVaR and apply gradient descent to solve for $u$. Hence, we proceed by first deriving the approximate value function for a fixed $u$ and time stage. Similar to POMDP whose value function can be represented by a set of so-called $\alpha$-functions, we first show an $\alpha$-function representation of the value function in BR-MDP, and then derive the approximate value function based on this representation. All proofs of the propositions/theorems below can be found in Appendix.

### 4.1 $\alpha$-function representation of the value function

By the definition of CVaR (see (1)) with a confidence level $\alpha$, we can rewrite the dynamic programming equation (5) for the BR-MDP as:

$$V_t^*(s_t, \mu_t) = \min_{\substack{a_t \in \mathcal{A} \\ u_t \in \mathbb{R}}} \left\{ u_t + \frac{1}{1 - \alpha} \int_\Theta \mu_t(\theta) \left( \int_\Xi f(\xi; \theta) \left( \mathcal{C}_t(s_t, a_t, \xi) + V_{t+1}^*(s_{t+1}, \mu_{t+1}) \right) d\xi - u_t \right)^+ d\theta \right\},$$
(6)

where we assume that $\xi$ and $\theta$ take continuous values, and the integration can be numerically approximated by Monte Carlo sampling. If $\xi$ and $\theta$ are discrete, the integral can be replaced by summation. The next proposition shows that the optimal value function corresponds to the lower envelope of a set of $\alpha$-functions.

**Proposition 4.1** ($\alpha$-function representation)**.** *The optimal value function in* (6) *can be represented by the lower envelope of a set of $\alpha$-functions denoted by $\Gamma_t = \{\alpha_t\}_{a_t \in \mathcal{A}}$, i.e.,*

$$V_t^*(s_t, \mu_t) = \min_{\alpha_t \in \Gamma_t} \int_\Theta \alpha_t(s_t, \theta) \mu_t(\theta) d\theta,$$

*where* $\alpha_t(s_t, \theta) = u_t + \frac{1}{1-\alpha} \left( \int_\Xi f(\xi; \theta) \left( \mathcal{C}_t(s_t, a_t, \xi) + \min_{\alpha_{t+1}} \int_\Theta \alpha_{t+1}(s_{t+1}, \theta) \frac{\mu_t(\theta) f(\xi; \theta)}{\int_\Theta \mu_t(\theta) f(\xi; \theta) d\theta} d\theta \right) d\xi - u_t \right)^+.$

Note that there is a major distinction between the $\alpha$-function representations of a POMDP and a BR-MDP: the optimal value function in the risk-neutral POMDP is piecewise linear and convex in the belief state (see [5]), whereas the optimal value function in BR-MDP is no longer piecewise linear in the posterior due to the $(\cdot)^+$ operator in CVaR. In addition, it is computationally impossible to obtain the $\alpha$-functions except for the last time stage. Specifically, denote the cardinality of the $\alpha_{t+1}$ set as $|\Gamma_{t+1}|$. Note that for each realization of $\xi$ there are $|\Gamma_{t+1}|$ candidates for the minimizer $\alpha_{t+1}^{*(\xi)}$, which attains the minimum of $\int_\Theta \alpha_{t+1}(s_{t+1}, \theta) \frac{\mu_t(\theta) f(\xi; \theta)}{\int_\Theta \mu_t(\theta) f(\xi; \theta) d\theta} d\theta$. There are a total of $|\mathcal{A}||\Gamma_{t+1}|^{|\Xi|}$ candidates for $\Gamma_t$, let alone the optimization over $u_t$. To deal with the difficulty in computing the $\alpha$-functions in POMDPs, [33] proposes several approximation algorithms, and later [34] extends the analysis to a continuous-state optimal stopping problem by applying Jensen's inequality to the exact value iteration in different ways and obtains a more computationally efficient approximation to the optimal value function. Inspired by these works, we derive an approximation approach next.

### 4.2 $\alpha$-function approximation

In this section, we derive the $\alpha$-function approximation for a fixed vector $(u_0, u_1, \cdots, u_{T-1})$. Without loss of generality, we assume the cost function at each time stage is non-negative (otherwise add a large constant to the cost at each time stage). For ease of exposition, we rewrite (6) as $V_t^*(s_t, \mu_t) = \min_{a_t \in \mathcal{A}, u_t \in \mathbb{R}} Q_t^*(s_t, \mu_t, a_t, u_t)$, where $Q_t^*(s_t, \mu_t, a_t, u_t) = u_t + \frac{1}{1-\alpha} \int_\Theta \mu_t(\theta) \left( \int_\Xi f(\xi; \theta) \left( \mathcal{C}_t(s_t, a_t, \xi) + V_{t+1}^*(s_{t+1}, \mu_{t+1}) \right) d\xi - u_t \right)^+ d\theta$. Let $V_t(s_t, \mu_t) := \min_{a_t} Q_t^*(s_t, \mu_t, a_t, u_t)$ be the "optimal" value function for a given $u_t$. Let $\underline{V}_t(s_t, \mu_t) := \min_{\underline{\alpha}_t \in \underline{\Gamma}_t} \int_\Theta \underline{\alpha}_t(s_t, \theta) \mu_t(\theta) d\theta$, where $\underline{\Gamma}_t = \{\underline{\alpha}_t\}_{a_t \in \mathcal{A}}$ and

$$\underline{\alpha}_t(s_t, \theta) = u_t + \frac{1}{1-\alpha} \int_\Xi \left( \mathcal{C}_t(s_t, a_t, \xi) f(\xi; \theta) - u_t + \min_{\alpha_{t+1}} \alpha_{t+1}(s_{t+1}, \theta) f(\xi; \theta) \right) d\xi.$$

$\underline{V}_t(s_t, \mu_t)$ serves as a lower bound for $V_t(s_t, \mu_t)$ (see Proposition 4.2), and is similar to the fast informed bound in POMDPs (see [33]). Note that the set $\underline{\Gamma}_t$ has a constant cardinality of $|\mathcal{A}|$. However, it involves a minimum within an integral, which can be hard to compute numerically. Also, the lower bound is loose in the sense that it could be negative, while the true CVaR value is always non-negative (due to the non-negative cost). Next, let $\bar{V}_t(s_t, \mu_t) := \min_{\bar{\alpha}_t \in \bar{\Gamma}_t} \int_\Theta \bar{\alpha}_t(s_t, \theta) \mu_t(\theta) d\theta$, where $\bar{\Gamma}_t = \{\bar{\alpha}_t\}_{a_t \in \mathcal{A}}$ and

$$\bar{\alpha}_t(s_t, \theta) = u_t + \frac{1}{1-\alpha} \left( \int_\Xi \mathcal{C}_t(s_t, a_t, \xi) f(\xi; \theta) d\xi - u_t \right)^+ + \frac{1}{1-\alpha} \int_\Xi \alpha_{t+1}(s_{t+1}, \theta) f(\xi; \theta) d\xi.$$

$\bar{V}_t(s_t, \mu_t)$ serves as an upper bound for $V_t(s_t, \mu_t)$ (see Proposition 4.2), and is similar to the unobservable MDP bound in POMDPs (see [33]), obtained by discarding all observations available to the decision maker. Suppose the cardinality at time stage $t+1$ is $|\bar{\Gamma}_{t+1}|$, then $\bar{\Gamma}_t$ has a total number of $|\mathcal{A}||\bar{\Gamma}_{t+1}|$ candidates. In the following, we derive another approximate value function $\widetilde{V}_t$ that is bounded by $\underline{V}_t$ and $\bar{V}_t$, and is at least better than one of the above bounds. It keeps a constant number $|\mathcal{A}|$ of $\alpha$-functions at each time stage, thus drastically reducing the computational complexity. Let $\widetilde{V}_t(s_t, \mu_t) := \min_{\widetilde{\alpha}_t \in \widetilde{\Gamma}_t} \int_\Theta \widetilde{\alpha}_t(s_t, \theta) \mu_t(\theta) d\theta$, where $\widetilde{\Gamma}_t = \{\widetilde{\alpha}_t\}_{a_t \in \mathcal{A}}$ and

$$\widetilde{\alpha}_t(s_t, \theta) = u_t + \frac{1}{1-\alpha} \left( \int_\Xi \mathcal{C}_t(s_t, a_t, \xi) f(\xi; \theta) d\xi - u_t + \min_{\alpha_{t+1}} \int_\Xi \alpha_{t+1}(s_{t+1}, \theta) f(\xi; \theta) d\xi \right)^+. \tag{7}$$

**Proposition 4.2.** *For all $t < T$ and any given $u_t \in \mathbb{R}$, the following inequalities hold:*

$$\underline{V}_t(s_t, \mu_t) \leq \widetilde{V}_t(s_t, \mu_t) \leq \bar{V}_t(s_t, \mu_t), \quad \underline{V}_t(s_t, \mu_t) \leq V_t(s_t, \mu_t) \leq \bar{V}_t(s_t, \mu_t).$$

To have an implementable algorithm, we need to use the approximate updating of $\alpha$-functions iteratively and replace the true $\alpha$-functions at the $t+1$ time stage in (7) by the approximate $\widetilde{\alpha}$-functions from the previous iteration. It is clear that the iterative approximations preserve the directions of the inequalities. Define $u_{t:} := (u_t, \cdots, u_{T-1})$. The approximate value function at time stage $t$ for a given $u_{t:}$ is given by $\widetilde{V}_t(s_t, \mu_t, u_{t:}) = \min_{\tilde{\alpha}_t \in \tilde{\Gamma}_t} \int_\Theta \widetilde{\alpha}_t(s_t, \theta) \mu_t(\theta) d\theta$, where $\tilde{\Gamma}_t = \{\tilde{\alpha}_t\}_{a_t \in \mathcal{A}}$ and

$$\widetilde{\alpha}_t(s_t, \theta) = u_t + \frac{1}{1-\alpha}\left(\int_\Xi \mathcal{C}_t(s_t, a_t, \xi)f(\xi; \theta)d\xi - u_t + \min_{\widetilde{\alpha}_{t+1}}\int_\Xi \widetilde{\alpha}_{t+1}(s_{t+1}, \theta)f(\xi; \theta)d\xi\right)^+. \quad (8)$$

## 4.3 Approximate dynamic programming with gradient descent

In this section, we incorporate $\alpha$-function approximation with gradient descent on $(u_0, u_1, \cdots, u_{T-1})$ based on the convexity of the approximate value function w.r.t. $(u_0, u_1, \cdots, u_{T-1})$, as formally shown in the theorem below.

**Theorem 4.3.** *Suppose the cost function $\mathcal{C}_t(s, a, \xi)$ is jointly convex in $(s, a)$ for any fixed $\xi$, and the state transition function $g_t(s, a, \xi)$ is jointly convex in $(s, a)$ for any fixed $\xi$. Then the approximate value function $\widetilde{V}_t(s_t, \mu_t, u_{t:})$ is convex in $u_{t:}$, for all $t < T$.*

The jointly convex assumption in Theorem 4.3 is common for gradient-based algorithms for solving multi-stage decision making problems (e.g. [35]). It is satisfied in many real-world problems such as inventory control (e.g. [36]) and portfolio optimization (e.g. [37]). We present the full algorithm in Algorithm 2.

---

**Algorithm 2:** Approximate dynamic programming for finite-horizon CVaR BR-MDPs.

---

**input**: finite horizon $T$, initial state $s_0$, prior distribution $\mu_0$, initial vector $u^0 = (u_0^0, u_1^0, \cdots, u_{T-1}^0)$, gradient descent step size $\eta_k$ for $k = 0, 1, \cdots$;
**initialization**: set $\widetilde{\alpha}_T(s_T, \theta) = \mathcal{C}_T(s_T), \forall s_T \in \mathcal{S}, \forall \theta \in \Theta$; set $k = 0$.
**while** *some stopping criterion is met* **do**
    **for** $t \leftarrow T - 1$ **to** *0* **do**
        | for each action $a_t \in \mathcal{A}$, compute $\widetilde{\alpha}_t(s_t, \theta)$ according to (8);
    **end**
    approximate the value function $\widetilde{V}_0(s_0, \mu_0, u^k) := \min_{\widetilde{\alpha}_0}\int_\Theta \widetilde{\alpha}_0(s_0, \theta)\mu_0(\theta)d\theta$;
    compute the gradient $\frac{\partial \widetilde{V}_0}{\partial u^k}$, update the vector $u^{k+1} = u^k - \eta_k \frac{\partial \widetilde{V}_0}{\partial u^k}$, set $k = k + 1$.
**end**
**output** the approximate value function $\widetilde{V}_0(s_0, \mu_0, u^k)$ and the optimal policy $\widetilde{\pi}_t(s_t, \mu_t) := \arg\min_{a_t \in \mathcal{A}}\int_\Theta \widetilde{\alpha}_t(s_t, a_t, \theta)\mu_t(\theta)d\theta.$

---

Note that in Algorithm 2, when applying the gradient descent, we need to compute the gradient of the approximate value function w.r.t. the vector $(u_0, u_1, \cdots, u_{T-1})$. For $\frac{\partial \widetilde{V}_0}{\partial u_0}$, we have

$$\frac{\partial \widetilde{V}_0}{\partial u_0} = \int_\Theta \left(1 - \frac{1}{1-\alpha}\mathbb{1}\left\{\int_\Xi \mathcal{C}_0(s_0, \widetilde{a}_0^*, \xi)f(\xi; \theta)d\xi - u_0 + \min_{\widetilde{\alpha}_1}\int_\Xi \widetilde{\alpha}_1(s_1, \theta)f(\xi; \theta)d\xi \geq 0\right\}\right)\mu_0(\theta)d\theta,$$

where $\widetilde{a}_0^* = \arg\min_{a_0 \in \mathcal{A}}\int_\Theta \widetilde{\alpha}_0(s_0, \theta)\mu_0(\theta)d\theta$. For $\frac{\partial \widetilde{V}_0}{\partial u_t}$, $t = 1, \cdots, T-1$, we have

$$\frac{\partial \widetilde{V}_0}{\partial u_t} = \int_\Theta \frac{1}{1-\alpha}\mathbb{1}\left\{\int_\Xi \mathcal{C}_0(s_0, \widetilde{a}_0^*, \xi)f(\xi; \theta)d\xi - u_0 + \min_{\widetilde{\alpha}_1}\int_\Xi \widetilde{\alpha}_1(s_1, \theta)f(\xi; \theta)d\xi \geq 0\right\}$$
$$\cdot \left[\int_\Xi \frac{\partial \widetilde{\alpha}_1}{\partial u_t}f(\xi; \theta)d\xi\right]\mu_0(\theta)d\theta,$$

where $\frac{\partial \widetilde{\alpha}_l}{\partial u_t}$ can be computed recursively from $\frac{\partial \widetilde{\alpha}_{l+1}}{\partial u_t}$ for $l = 1, \cdots, t-1$.

The approximate value function output by Algorithm 2 provides an upper bound on the optimal value function, which is shown in the theorem below.

**Theorem 4.4.** $\min_{u_{t:}} \widetilde{V}_t(s_t, \mu_t, u_{t:})$ *is an upper bound for the optimal value function $V_t^*(s_t, \mu_t)$.*

Even though the approximate value function is an upper bound on the optimal value function, we will later show in the numerical experiments that the gap between these two is small. As a final note, even though we develop the algorithm for the risk functional CVaR, it can be extended easily to other coherent risk measures. Consider a class of coherent risk measures which can be represented in the

following parametric form $\mathcal{R}(Z) := \inf_{\lambda \in \Lambda} \mathbb{E}[\Psi(Z, \lambda)]$, where $\Psi : \mathbb{R} \times \Lambda \to \mathbb{R}$ is a real-valued function and $\Psi(z, \lambda)$ is convex in $(z, \lambda)$. CVaR is an example of such coherent risk measure. As another example, consider the following coherent risk measure based on Kullback–Leibler divergence [38], which is also an example given in [35]. Here the risk functional takes the form

$$\mathcal{R}_\epsilon(Z) = \inf_{\gamma, \lambda > 0} \left\{ \lambda\epsilon + \gamma + \lambda e^{-\gamma/\lambda} \mathbb{E}\left[e^{Z/\lambda}\right] - \lambda \right\},$$

where $\epsilon$ is the user-defined ambiguity set size. It can be easily checked that this coherent risk measure takes the required form $\mathcal{R}(Z) := \inf_{\lambda \in \Lambda} \mathbb{E}[\Psi(Z, \lambda)]$, for some $\lambda$ and $\Psi$ function. We can write the dynamic programming equation for the BR-MDP with the above coherent risk measure as:

$$V_t^*(s_t, \mu_t) = \min_{\substack{a_t \in \mathcal{A} \\ \lambda_t \in \Lambda}} \left\{ \int_\Theta \mu_t(\theta) \Psi(\int_\Xi f(\xi; \theta) \left(\mathcal{C}_t(s_t, a_t, \xi) + V_{t+1}^*(s_{t+1}, \mu_{t+1})\right) d\xi, \lambda_t) d\theta \right\}.$$

Following the same procedure, we can use $\alpha$-function to represent the value function, apply the same technique to approximate the $\alpha$-functions for a given vector $\lambda_0, \cdots, \lambda_{T-1}$, and then apply gradient descent on the approximate value function. The convergence is guaranteed due to the convexity. Since the derivation of $\alpha$-function representation and approximation (obtained by applying Jensen's inequality) are essentially the same, we omit the full procedures.

## 5 Numerical experiments

We illustrate the performance of our proposed formulation and algorithms with two **offline planning** problems.

(1) **Gambler's betting problem**. Consider a gambler betting in the casino with initial money of $s_0$. At each time stage the gambler chooses how much to bet from a set $\{0, 1, 2, 3, 5\}$. The gambler bets for $T = 6$ rounds. The cost at each time stage is $-a \cdot \xi$, where $a$ stands for action of how much to bet, $\xi = 2$ stands for a win, $\xi = -1$ stands for a loss, and the winning rate $\theta^c = \mathbb{P}(\xi = 2)$ is unknown. We add a constant $c = 10$ to make the adjusted cost $c - a \cdot \xi$ non-negative to run our algorithm (since the algorithm requires non-negative stage-wise cost), and then subtract $cT$ from the resultant total adjusted cost to recover the total cost. The data set consists of historical betting records with size $N$.

(2) **Inventory control problem**. Consider a warehouse manager with initial inventory level $s_0$. At each time stage the manager chooses how much to replenish from the set $\{0, 1, \cdots, M - s\}$, where $M = 15$ is the storage capacity. The customer demand is a random variable $\xi$ following a Poisson distribution with parameter $\theta^c$ truncated below $M_C = 20$, which is the maximal customer demand the warehouse can handle. The state transition is given by $s_{t+1} = \max(s_t + a_t - \xi_t, 0)$, the cost function is given by $\mathcal{C}_t(s_t, a_t, \xi_t) = h_t \cdot \max(s_t + a_t - \xi_t, 0) + p_t \cdot \max(\xi_t - s_t - a_t, 0)$, where $h_t$ is the holding cost and $p_t$ is the penalty cost. The final stage cost is set to 0 for simplicity. The manager has to plan for $T = 6$ time stages. The data set consists of historical customer demands with size $N$.

We compare the following approaches.

(1) BR-MDP (e.): exact dynamic programming (Algorithm 1), where the state $s_t$ and $\mu_t$ are discretized to a fine grid to carry out the dynamic programming (see Appendix for details).

(2) BR-MDP (a.): approximate dynamic programming (Algorithm 2).

(3) Nominal: maximal likelihood estimation (MLE) estimator $\theta_{\text{MLE}}$ is computed from the given data, and then a policy is obtained by solving the MDP with parameter $\theta_{\text{MLE}}$.

(4) DR-MDP: distributionally robust MDP presented in [1].

For each of the considered approaches, we obtain the corresponding optimal policy for a same data set, and then evaluate the actual performance of the obtained policy on the true system, i.e., MDP with the true parameter $\theta^c$. This is referred to as one replication, and we repeat the experiments for 100 replications on different independent data sets. Results for the gambler's betting problem can be found in Table 1a, Table 2, Figure 1a and Figure 1b. Results for the inventory control problem can be found in Table 1b.

Table 1: Average time to solve each formulation, mean and variance of actual performance of the solved policy on 100 replications. Data size $N = 10$.

| Approach | $\theta^c = 0.45$ | | | $\theta^c = 0.55$ | | $\theta^c = 12$ | | |
|---|---|---|---|---|---|---|---|---|
| | time(s) | mean | variance | mean | variance | time(s) | mean | variance |
| BR-MDP (e., $\alpha = 0.4$) | 65.52 | -8.82 | 9.92 | -17.83 | 8.24 | 2951.12 | 81.63 | 5.15 |
| BR-MDP (a., $\alpha = 0.4$) | 5.02 | -8.26 | 11.42 | -17.16 | 6.50 | 224.57 | 83.55 | 12.82 |
| Nominal | 0.74 | -6.30 | 26.46 | -17.95 | 34.22 | 2.58 | 84.44 | 54.17 |
| BR-MDP (e., $\alpha = 1$) | 67.83 | -2.38 | 7.02 | -4.25 | 4.49 | 2947.20 | 83.25 | 3.46 |
| DR-MDP | 0.69 | 0.00 | 0.00 | 0.00 | 0.00 | 2.44 | 99.77 | 0.00 |

(a) Betting problem.  (b) Inventory problem.

Table 2: Mean and variance of actual performance in the betting problem. Data size $N$ varies from 5, 10, to 100. Experiments are run on 100 replications.

| Approach | $\theta^c = 0.45$ | | | | | | $\theta^c = 0.55$ | | | | | |
|---|---|---|---|---|---|---|---|---|---|---|---|---|
| | $N = 5$ | | $N = 10$ | | $N = 100$ | | $N = 5$ | | $N = 10$ | | $N = 100$ | |
| | mean | variance | mean | variance | mean | variance | mean | variance | mean | variance | mean | variance |
| BR-MDP (e., $\alpha = 0.4$) | -7.83 | 14.67 | -8.82 | 9.92 | -9.26 | 7.51 | -16.27 | 15.05 | -17.83 | 8.24 | -18.12 | 5.90 |
| BR-MDP (a., $\alpha = 0.4$) | -7.21 | 15.44 | -8.26 | 11.42 | -9.13 | 7.73 | -16.12 | 15.52 | -17.16 | 6.50 | -17.89 | 6.20 |
| Nominal | -5.88 | 54.12 | -6.30 | 26.46 | -9.45 | 9.92 | -15.85 | 46.92 | -17.95 | 34.22 | -18.25 | 6.92 |
| BR-MDP (e., $\alpha = 1$) | -2.12 | 6.66 | -2.38 | 7.02 | -1.15 | 3.42 | -3.65 | 8.35 | -4.25 | 4.49 | -1.36 | 2.46 |
| DR-MDP | -0.10 | 1.09 | 0.00 | 0.00 | 0.00 | 0.00 | -0.12 | 1.31 | 0.00 | 0.00 | 0.00 | 0.00 |

Table 1 reports the average computation time to obtain the optimal policy and the mean and variance of the actual performance of the obtained policy over the 100 replications. Table 2 reports the actual performance of different formulations over 100 replications with different data size $N = 5, 10, 100$. We have the following observations.

(1) **Robustness of BR-MDP**: BR-MDP is the most robust in the sense of balancing the mean (smaller mean cost) and variability (smaller variance) of the actual performance of its solution. In contrast, the nominal approach has much larger variance, especially when the data size is small, indicating it is not robust against parameter uncertainty. On the other hand, DR-MDP is overly conservative since the variance of actual performance is 0 and the mean is the largest (i.e., the worst) among all approaches. This conservativeness is often not desirable: for example, in the betting problem, DR-MDP always chooses the conservative action "not bet", which is obviously not optimal when $\theta^c$ (probability of winning) is large and the goal is to minimize the expected cost.

(2) **Efficiency of the approximation algorithm for BR-MDP**: the computation time of the approximation algorithm for BR-MDP is less than 1/10 of that of the exact algorithm, while the performance (mean, variance) is not much different.

(3) **Larger data size reduces parameter uncertainty**: as expected, as we have more data, the uncertainty about model parameters reduces. Hence, the benefit of considering future data realization in BR-MDP decreases compared to the nominal approach, resulting in their similar performance when the data size is large ($N = 100$ in our examples). The reason is that both the posterior distribution (used in BR-MDP) and the MLE estimator (used in the nominal approach ) converge to the true parameter as data size goes to infinity.

Figure 1a shows the histogram of actual performance over 100 replications for the nominal approach and CVaR BR-MDP (exact) with different confidence levels $\alpha = 0.1, 0.5, 0.99$ under $\theta^c = 0.45$. We have the following observations combining Table 1 and Figure 1a:

(1) **Robustness of BR-MDP**: BR-MDP (both exact and approximate) produce more consistent solution performance across a wide range of input data compared to the nominal approach, which can be seen from the smaller variance in Table 1 and more concentrated distribution of the actual performance in Figure 1a.

(2) **Benefit of learning from future data realization (time consistency)**: Figure 1a shows that in the betting problem with $\theta^c = 0.45$, the nominal approach has 40 replications where MLE estimator $\theta_{\mathrm{MLE}} < 0.33$ and the gambler chooses not to bet, which is not the optimal action. In contrast, BR-MDP formulation learns from the future data realization and updates its posterior distribution on $\theta$. As a result, in those 40 replications, the gambler initially chooses not to bet, but after some time chooses to bet, which results in the left-shift of

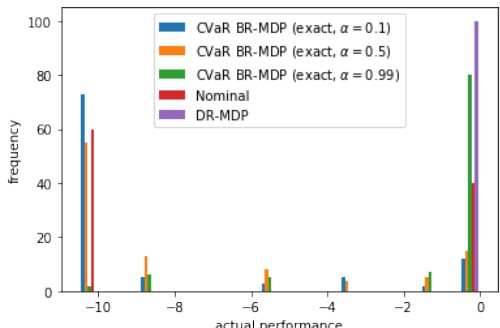

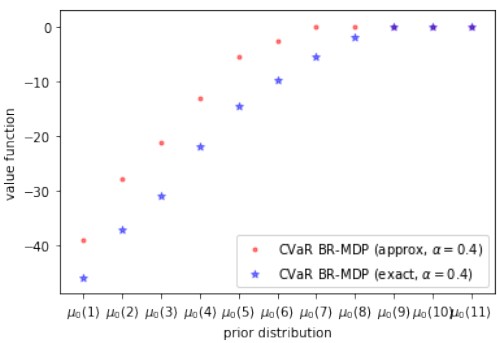

(a) Histogram of actual performance over 100 replications for CVaR BR-MDP (exact) with different $\alpha$.

(b) Value functions of CVaR BR-MDP (exact and approx) under different priors.

the actual performance distribution. This illustrates time consistency (or in other words, adaptivity to the data process) of our BR-MDP formulation. This illustration is even more evident by the comparison between DR-MDP and BR-MDP with $\alpha = 1$ (CVaR with $\alpha = 1$ corresponds to the worst-case measure), where the only difference is that BR-MDP takes a nested form of risk functional while DR-MDP uses a static one.

(3) **Effect of risk level**: risk level $\alpha$ affects the conservativeness of BR-MDP; as $\alpha$ increases, the gambler is more likely to take a conservative action (which is not to bet), so the actual performance distribution will shift more to the right.

(4) **Effectiveness of the approximation algorithm for BR-MDP**: Figure 1b plots the value function $V_0^*(s_0, \mu_0)$ of BR-MDP (exact) and $\tilde{V}_0^*(s_0, \mu_0)$ of BR-MDP (approx) under different prior distributions $\mu_0$ with $\theta^c = 0.45$, verifying Theorem 4.4 that $\tilde{V}_0^*(s_0, \mu_0)$ is indeed an upper bound for $V_0^*(s_0, \mu_0)$ but the difference between these two is small.

## 6 Conclusion

In this paper, we propose a new formulation, coined as Bayesian Risk MDP (BR-MDP), to provide robustness against parameter uncertainty in MDPs. BR-MDP is a time-consistent formulation with a dynamic risk functional that seeks the trade-off between the posterior expected performance and the robustness in the actual performance. For finite-horizon BR-MDP with the CVaR risk functional, we develop an efficient approximation algorithm by drawing a connection between BR-MDPs and POMDPs and deriving an approximate alpha-function representation that remains a low computational cost. Our experiment results demonstrate the efficiency of the proposed approximate algorithm, and show the robustness and the adaptivity to future data realization of the BR-MDP formulation.

One of the limitations of our work is the parametric assumption on the distribution of randomness. In the future work, we wish to extend the BR-MDP formulation to non-parametric Bayesian setting, and evaluate the performance of the proposed formulation and algorithm on real-world data sets in more challenging problems. In addition, the proposed alpha-function approximation algorithm provides an upper bound of the exact value, while there is no theoretical guarantee on the gap between the two. In future we will develop more efficient approximation algorithms with a convergence guarantee, such as methods based on stochastic dual dynamic programming. There are also other interesting directions, such as extending the BR-MDP formulation to an infinite horizon problem and utilizing function approximation to improve the scalability of the proposed approach to more complex domains.

## Acknowledgments and Disclosure of Funding

The authors gratefully acknowledge the support by the Air Force Office of Scientific Research under Grant FA9550-19-1-0283 and Grant FA9550-22-1-0244, and National Science Foundation under Grant DMS2053489.

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
