# OpenReview forum: "Bayesian Risk Markov Decision Processes"
_NeurIPS.cc/2022/Conference — NeurIPS 2022 Accept_

### Official Review · Reviewer_TzUd · 2022-07-06

**Rating:** 6
**Confidence:** 3
**Soundness:** 4 excellent
**Presentation:** 3 good
**Contribution:** 2 fair

**Summary:**

This paper proposes the Bayesian Risk MDP (BR-MDP) framework to deal with the epistemic uncertainty in sequential decision problems. In the introduced framework, a coherent risk functional is applied to the expectation of the total cost under the posterior distribution of the model parameters. Differently from prior works, the risk functional is applied in a nested form, which preserves the time consistency of the objective and allows to write a dynamic programming equation of the optimal value function. However, solving the dynamic programming problem analytically is far-fetched. The paper thus introduces an approximate algorithm, partially inspired by POMDP methods, for a specific instance of the BR-MDP with the CVaR risk functional. Finally, the given methodology is empirically evaluated in toy domains.

**Questions:**

1) The work by Chow et al., 2015 [22] seems to be closely related to this submission. Although they tackle the cost-variability risk instead of the epistemic risk, they also consider a CVaR risk measure in a nested form to derive a Bellman equation (Sec. 3) that allows for approximate dynamic programming. Can the authors discuss the relation with this prior work, and especially the technical differences with respect to this submission?

2) Chow et al., 2015 [22] make an interesting connection between cost-variability risk and epistemic risk. They prove that a solution that is sensitive to the cost-variability provides also robustness to the epistemic uncertainty as a by-product. Do the authors think that a similar case could be made in their setting, i.e., that a solution to the BR-MDP problem could provide some robustness to the cost-variability risk as well?

3) The proposed BR-MDP framework is similar in flavor to a Bayes-adaptive approach with a time consistent risk functional over the model parameters. Can the authors discuss the impact of the time consistency? Can they explain why an approximate solution to the BR-MDP problem is necessarily better than a (possibly exact) solution to the BAMDP problem with a risk functional applied on the full trajectory rather than in a nested form?

4) Is the proposed approximate methodology really practical? Can the authors comment on how the Eq. 8 could be computed/estimated in more challenging domains?

5) Can the authors better explain why the Eq. 7 is significantly easier to compute than the exact Eq. 6? I guess the main benefit is pulling the min out of the expectation, but I found this paragraph quite hard to process (especially lines 218-222 could be revised).

6) Theorem 3.4 characterizes the approximate value function as an upper bound of the exact value. Do the authors believe there is hope to assess guarantees on the gap between the two?

**Limitations:**

I think that a discussion on the negative societal impact can be avoided in this paper. However, the authors answered 'Yes' to the checklist question on the limitations of their work, but they did not provide motivation or context for their answer. Can the authors list the main limitations of their approach?

**Strengths And Weaknesses:**

Strengths
- (Relevant Problem) This paper tackles the relevant problem of robust decision-making under uncertainty on the model parameters.
- (Framework) I am not particularly familiar with the related literature, but to my understanding this is the first Bayes-adaptive framework that incorporates a time consistent risk measure over the epistemic uncertainty.
- (Methodology) Since solving the BR-MDP problem is intractable in general, the paper proposes an interesting value function approximation method that is partially inspired by POMDP literature.

Weaknesses
- (Weak Empirical Analysis) The empirical analysis does not fully motivate the framework and the methodology, as the performance of the proposed approach does not seem to be significantly superior than the (not necessarily strong) baselines.
- (Notation and Clarity) The notation is quite convoluted and not always sharp. This undermines the clarity of some portions of the work.

The premise of incorporating the sensitivity to the epistemic risk in a Bayes-adaptive framework is interesting, and also a natural continuation of the robust decision-making stream of works. The technical contribution does not seem to be ground-breaking, as most of the ideas can be traced back to previous works in cost-variability sensitivity and POMDPs, but this would be totally fine given the other contributions, i.e., the methodology and the framework itself. My main concern regards the empirical validation: It is ok for an essentially methodological/theoretical paper to have an empirical analysis in toy problems, but it should showcase the benefit brought by the method. Instead, the performance seems to be close to the baselines, especially to the naïve maximum likelihood estimator.

For the aforementioned reasons, I am still unsure on the significance of the work, and I am providing a slightly negative evaluation. However, I am open to increase my score if the authors can address my concerns on the motivation and potential of the proposed approach in their rebuttal.

---
AFTER DISCUSSION

I have increased my score to weak accept to acknowledge the value of this work that emerged from discussion with the authors and other reviewers. However, I believe this paper is still missing either a stronger theoretical result on the benefit of the nested form or a stronger empirical evaluation in more challenging domains to be outstanding in any way.

---

> ### Author Response · Authors · 2022-08-02
> **Response to Reviewer TzUd I**
>
> Thank you very much for the valuable time you have spent reviewing our work. Below are our responses to the comments and questions you have.
>
> - **My main concern regards the empirical validation: It is ok for an essentially methodological/theoretical paper to have an empirical analysis in toy problems, but it should showcase the benefit brought by the method. Instead, the performance seems to be close to the baselines, especially to the naive maximum likelihood estimator**.
>
> Thank you for raising this concern. We found we made a mistake in our previous implementation: we added a constant $c$ (which is an upper bound on stage-wise constant) to the cost at each time stage to ensure the stage-wise cost non-negative (as required by our algorithm), but forgot to subtract $cT$ (which is the total added constant over $T$ horizons) from the algorithm output to recover the total cost of the problem. As a result, the added constant $cT$ dominates the total cost and obscures the difference between different methods. After correcting this mistake, now we can see more difference between different formulations and also see the benefits of BR-MDP: it balances the trade-off between the mean and variance of the actual performance of its solution. In particular, we can see the nominal approach often results in the largest variance in actual performance, indicating it does not provide any robustness against distributional shift. On the hand, DR-MDP often results in the most conservative policy that has 0 variance in its actual performance but the mean cost is usually the worst. BR-MDP strikes the middle ground between these two and provide a good balance between mean performance and variability of the actual performance. For more details, please see the numerical section in the revised version for more details.
>
> - **The notation is quite convoluted and not always sharp. This undermines the clarity of some portions of the work**.
>
> Thank you for your comment. The heavy notations are due to the complexity of the considered problem (risk-averse settings, CVaR, alpha function approximation etc.). In this revised version, we have tried to condense and streamline the notations as much as possible, and also provided a table summarizing main notations in the appendix for future reference.
>
> - **The work by Chow et al., 2015 [1] seems to be closely related to this submission. Although they tackle the cost-variability risk instead of the epistemic risk, they also consider a CVaR risk measure in a nested form to derive a Bellman equation (Sec. 3) that allows for approximate dynamic programming. Can the authors discuss the relation with this prior work, and especially the technical differences with respect to this submission**.
>
> Thank you for your comment. There are two key differences between work [1] and our submission. First, [1] considers a standard MDPs with CVaR replacing expectation to account for the aleatoric uncertainty. Cost function and transition probabilities are both known in their case. We consider the case where we don't know those cost function and transition probabilities, and the CVaR risk functional is used to account for the epistemic uncertainty. Second, due to the unknown nature of the MDP problem in our setting, as we take an action and observe the state transition, we will update the posterior distribution on the unknown parameter. This Bayesian updating results in an exponential growth in the set of reachable augmented states, which prohibits the use of the value iteration approach proposed in [1].
>
> [1] Chow, Y., Tamar, A., Mannor S., and Pavone M., 2015. Risk-sensitive and Robust Decision-making: a
> CVaR Optimization Approach. Advances in Neural Information Processing Systems, 28.

---

> ### Author Response · Authors · 2022-08-02
> **Response to Reviewer TzUd II**
>
> - **Chow et al., 2015 [1] make an interesting connection between cost-variability risk and epistemic risk. They prove that a solution that is sensitive to the cost-variability provides also robustness to the epistemic uncertainty as a by-product. Do the authors think that a similar case could be made in their setting, i.e., that a solution to the BR-MDP problem could provide some robustness to the cost-variability risk as well**.
>
> Thank you for bringing up this interesting point. We do not have a definitive answer to this question, but our conjecture is that BR-MDP provides robustness in a similar sense as BRO (in static optimization), which balances trade-off between the posterior mean performance and the variability of actual performance of its solution. However, the DRO approach (proposed for hedging against the epistemic uncertainty in static optimization) has been show to provide robustness to the cost-variability risk when the ambiguity set is small, see [2][3]. So, DRO probably has a close connection with [1]. As far as we know, the interpretations of robustness in DRO or BRO have only been rigorously studied for static optimization, and it would be an interesting direction to study if these interpretations carry over to the dynamic problem.
>
> - **The proposed BR-MDP framework is similar in flavor to a Bayes-adaptive approach with a time consistent risk functional over the model parameters. Can the authors discuss the impact of the time consistency? Can they explain why an approximate solution to the BR-MDP problem is necessarily better than a (possibly exact) solution to the BAMDP problem with a risk functional applied on the full trajectory rather than in a nested form**.
>
> Thank you for your comment. The impact of time-consistency has been discussed extensively in the literature (see [4], [5]). One of the considerations for time-consistency in this paper is its ``dynamic programming'' style property: for a chosen risk functional, if a policy is risk-optimal for an $T$-stage problem, then the component of the policy from the $t^{th}$ time until the end (where $t < T$) is also risk-optimal. The nested form can be show analytically better than the static form (i.e., the risk functional applied to the full trajectory); please see appendix for the proof on a three-stage problem, which can be extended to any number of stages. Here, we illustrate it with the betting problem considered in this paper. Consider a two-stage betting problem, where we are only given five past betting records with four wins and one loss, and the true winning rate is $10\%$. The risk functional is chosen to be CVaR with $\alpha=\frac{1}{5}$. A win yields a cost of -2 and a loss yields a cost of 1. The gambler decides to bet or not for two runs. It can be easily checked that the optimal policy for the static formulation is always to bet in the two runs. The optimal policy for the nested formulation is to bet in the first run. If it turns out to be a win, then the gambler bets in the second, otherwise chooses not to bet. When evaluating the two policies on the true model (winning rate is $10\%$), one could easily see the performance of the optimal policy in the nested formulation is better than that in the static formulation.
>
> - **Is the proposed approximate methodology really practical? Can the authors comment on how the Eq. 8 could be computed/estimated in more challenging domains**.
>
> Thank you for your question. For more challenging domains, one could expect longer horizons, larger state space and action space, higher dimension of parameter space and randomness space, and thus high-dimensional integration may be involved. The proposed approximated methodology remains practical in this case. Note that the number of alpha-functions at each time stage is constant and equals the cardinality of the action space. For high-dimensional integration, one can resort to Monte Carlo integration which enjoys a convergence rate of $1/\sqrt{\text{sample size}}$ and in independent of the dimension. In the future work, we wish to utilize function approximation to improve the scalability of our approach to more complex domains.
>
> [2]  Jun-ya Gotoh, Michael Jong Kim, and Andrew Lim. Robust Empirical Optimization is Almost
> the Same as Mean-variance Optimization. Operations Research Letters, 46(4):448–452, 2018.
>
> [3] John C. Duchi, Peter W. Glynn, and Hongseok Namkoong. Statistics of Robust Optimization: A Generalized Empirical Likelihood Approach. Mathematics of Operations Research,
> 46(3):946–969, 2021.
>
> [4] Iancu, D., Petrik M., and Subramanian D., 2015. Tight Approximations of Dynamic Risk Measures. Mathematics of Operations Research, 40(3):655–682.
>
> [5] Shapiro, A., 2021. Tutorial on Risk Neutral, Distributionally Robust and Risk Averse Multistage Stochastic Programming. European Journal of Operational Research, 288(1), pp.1-13.

---

> ### Author Response · Authors · 2022-08-02
> **Response to Reviewer TzUd III**
>
> - **Can the authors better explain why the Eq. 7 is significantly easier to compute than the exact Eq. 6? I guess the main benefit is pulling the min out of the expectation, but I found this paragraph quite hard to process (especially lines 218-222 could be revised)**.
>
> Thank you for your comment. Please see line 200-203 for a revised statement. For the exact alpha-function representation like in Eq. 6, to compute $\alpha_t(s_t,\theta)$, we need to find the minimizer $\alpha_{t+1}^{\*(\xi)}$, which attains the minimum of $\int_{\Theta} \alpha_{t+1}(s_{t+1}, \theta) \frac{\mu_t(\theta)f(\xi;\theta)}{\int_{\Theta}\mu_t(\theta)f(\xi;\theta)d\theta}d\theta$ for every realization of $\xi$. We use superscript $\xi$ to explicitly show that for each $\xi$, we need to find a minimizer $\alpha_{t+1}^{\*(\xi)}$ in set $\Gamma_{t+1}$. Since for each $\xi$, there are $|\Gamma_{t+1}|$ candidates for $\alpha_{t+1}^{*(\xi)}$ (which means one has to search over those candidates to find the minimizer), there are a total of $|\Gamma_{t+1}|^{|\Xi|}$ candidates for the alpha function at time stage $t$. On the other hand, in Eq. 7, the minimum is outside the integral over $\xi$, such that we do not need to find minimizer for each $\xi$, thus we greatly reduce the number of candidates for the alpha functions.
>
> - **Theorem 3.4 characterizes the approximate value function as an upper bound of the exact value. Do the authors believe there is hope to assess guarantees on the gap between the two**.
>
> Thank you for your comment. For the proposed alpha function approximation algorithm, there is no theoretical guarantee on the gap between the two. Similar to [6] and [7], we apply Jensen's inequality to exchange the order of minimum and integral, and the gap between the two will increase as the time horizon increases. Hence, this class of approaches are more suitable for problems with a small time horizon.
>
> - **The authors answered 'Yes' to the checklist question on the limitations of their work, but they did not provide motivation or context for their answer. Can the authors list the main limitations of their approach**.
>
> Thank you for your comment. We have added the following discussion of limitations of our approach in the conclusion section in our revised version:
>
> "One of the limitations of our work is the parametric assumption on the distribution of randomness. In the future work, we wish to extend the BR-MDP formulation to non-parametric Bayesian setting, and evaluate the performance of the proposed formulation and algorithm on real-world data sets in more challenging problems. In addition, the proposed alpha-function approximation algorithm provides an upper bound of the exact value, while there is no theoretical guarantee on the gap between the two. In future we will develop more efficient approximation algorithms with a convergence guarantee, such as methods based on stochastic dual dynamic programming. There are also other interesting directions, such as extending the BR-MDP formulation to an infinite horizon problem and utilizing function approximation to improve the scalability of the proposed approach to more complex domains."
>
> [6] Hauskrecht, M., 2000. Value-function Approximations for Partially Observable Markov Decision Processes. Journal of artificial intelligence research, 13:33–94.
>
> [7] Zhou, E., 2013. Optimal Stopping under Partial Observation: Near-value Iteration. IEEE Transactions on Automatic Control, 58(2):500–506.

---

> > ### Comment · Reviewer_TzUd · 2022-08-06
> > **Follow-up to Response**
> >
> > First, I want to thank the authors for their thorough replies. I would suggest them to clearly mark the changes made to the manuscript (e.g., reporting the new parts in a different color) so that it will be easier for reviewers to go through them.
> >
> > I still have a couple of questions regarding the significance of time-consistency and the experimental results.
> >
> > 1) I got the point that the nested (time-consistent) form is analytically better than the static form. However, the nested form can only be solved approximately, whereas the static form is amenable for computation. Can we say that the approximated nested form is still analytically better than the static form? I believe this point would require further discussion, as it is crucial to motivate the approach in the first place.
> >
> > 2) I have briefly looked at the updated numerical results: Even after filtering out the $cT$ constant, it is still not clear to me how can we conclude that BR-MDP is significantly better. Can the authors better explain what to look for in the experimental results, and how they are supporting the theoretical claims in the paper?

---

> > > ### Author Response · Authors · 2022-08-08
> > > **Follow-up response to Reviewer TzUd**
> > >
> > > Thank you very much for your valuable suggestion. We have marked the main changes in the revised version in red. Below are our responses to your latest comments and questions.
> > >
> > > -- **I got the point that the nested (time-consistent) form is analytically better than the static form. However, the nested form can only be solved approximately, whereas the static form is amenable for computation. Can we say that the approximated nested form is still analytically better than the static form? I believe this point would require further discussion, as it is crucial to motivate the approach in the first place.**
> > >
> > > Thank you for your question. First, the static form can only be solved approximately as well, since it also involves additional continuous state. For example, [1] considers the static CVaR risk functional and only shows an approximation algorithm to the value iteration due to the introduction of an additional continuous state; [2] also optimizes a static CVaR risk functional over the total cost and proposes some approximation algorithm. Second, since the static form can only be solved approximately as well, it would be unfair to compare the *approximated* nested form with the *exact* static form, while it is fair to say the *exact* nested form is analytically better than the *exact* static form.
> > >
> > > -- **I have briefly looked at the updated numerical results: Even after filtering out the  constant, it is still not clear to me how can we conclude that BR-MDP is significantly better. Can the authors better explain what to look for in the experimental results, and how they are supporting the theoretical claims in the paper?**
> > >
> > > Thank you for your question. The main thing to look for in the experimental results is the balance of the mean and variance of the actual performance of the each formulation's solution: the smaller mean cost and the smaller variance, the better the approach. In a nutshell, BR-MDP achieves the best balance between the mean and the variance; whereas the nominal approach has a much larger variance (indicating no robustness, subject to distributional shift), and DR-MDP has 0 variance but worst mean (indicating although it is robust but overly conservative).  We have revised the explanations in the numerical section to make it more clear.
> > >
> > > In this latest revised version, we also made a slight change to the betting problem setting as follows: when betting one dollar, a win results in two dollars (previously a win was one dollar); everything else stays the same. This change enlarges the range of expect cost values, and hence makes the differences among different approaches more obvious. We report the new numerical result in Table 1 and Table 2 in this revised version. For example, when the data size is small ($N=5$), our proposed BR-MDP formulation provides more robustness (much smaller variance) than the nominal approach. This can be seen from variance $14.67$ of BR-MDP formulation versus $54.12$ of nominal approach when $\theta^c=0.45$, and $15.05$ of BR-MDP formulation versus $46.92$ of nominal approach when $\theta^c=0.55$. In terms of the mean cost, our BR-MDP formulation is also better compared to the nominal approach when the data size is small. This can be seen from mean $-7.83$ of BR-MDP formulation versus $-5.88$ of nominal approach when $\theta^c=0.45$, and $-16.27$ of BR-MDP formulation versus $-15.85$ of nominal approach when $\theta^c=0.55$. One of the reasons is the adaptivity to the data process of our BR-MDP formulation. When the data size is small, the nominal approach is subject to the parameter uncertainty, and the plugged-in MLE estimator may deviate from its true value a lot. On the other hand, our BR-MDP formulation takes into consideration the future data realization, and thus produce more consistent policy over different replications that also behaves well under the true model. Finally, compared to DR-MDP which only fixates on the worst-case scenario, our BR-MDP formulation is much less conservative. This can be seen from the much better mean cost of our BR-MDP formulation than the DR-MDP formulation.
> > >
> > >
> > > [1] Chow, Y., Tamar, A., Mannor S., and Pavone M., 2015. Risk-sensitive and Robust Decision-making: a CVaR Optimization Approach. Advances in Neural Information Processing Systems, 28.
> > >
> > > [2] Rigter, M., Lacerda, B. and Hawes, N., 2021. Risk-averse Bayes-adaptive Reinforcement Learning. Advances in Neural Information Processing Systems, 34, pp.1142-1154.

---

> > > > ### Comment · Reviewer_TzUd · 2022-08-09
> > > > **Thanks for the clarifications**
> > > >
> > > > I want to thank the authors for their further clarifications over my follow-up comments. I believe I now have a better understanding of the merits of this submission, and I will discuss them with other reviewers before providing a final recommendation.

---

> > > > > ### Author Response · Authors · 2022-08-09
> > > > > **Response to Reviewer TzUd**
> > > > >
> > > > > Thank you for your reply and acknowledgement! We look forward to your final recommendation!

---

### Official Review · Reviewer_sCox · 2022-07-07

**Rating:** 6
**Confidence:** 4
**Soundness:** 3 good
**Presentation:** 3 good
**Contribution:** 2 fair

**Summary:**

This paper proposes a Bayesian risk MDP called BR-MDP to account for both the uncertainties in transition probabilities and costs/rewards. The model takes a nested form of the risk functional, which endows its optimal policy with time consistency. The computational difficulties in the expectation and the risk functional, as well as the posterior with a possibly infinite dimension, are two main challenges in obtaining the solution of the BR-MDP. To overcome the latter, the authors focus on the conjugate families of distributions, while for the former, the authors consider the conditional value-at-risk and derive an $\alpha$-function approximation of the optimal value function and propose an algorithm to solve for the approximate value function efficiently. Two empirical studies are carried out to demonstrate the performance of the proposed model and the efficiency of the proposed algorithm.

**Questions:**


Below are some other comments for the paper.

$\bullet$ Line $138$, page $4$ (Section $2$): I wonder what is the meaning of $\Xi_t$. Is it a random variable or a set?

$\bullet$ Line $164$, page $4$ (Section $2$): by "To illustrate", I expect the authors to demonstrate the time consistency of the BR-MDP, but they just show that the objective value of BR-MDP serves as an upper bound for the objective of the one considering static risk functional. I would suggest the authors relate this to the concept of time consistency.

$\bullet$ Line $166$, page $4$ (Section $2$): based on formula $(2)$, should it be $\mathcal{C}_1(s_1)$ rather than $\mathcal{C}_1(s_1,a_1,\xi_1)$?

$\bullet$ Line $212$, page $6$ (Section $3$): the definition $\Gamma_t=\{\alpha_t\}_{a_t\in\mathcal{A}}$ is confusing. Does it imply that the cardinalities of $\Gamma_0, \Gamma_1,...$ are all $\vert\mathcal{A}\vert$?

$\bullet$ Line $294$, page $8$ (Section $4$): what is the value of $M_C$?

$\bullet$ Lines $312$ to $313$, page $8$ (Section $4$): it seems to me that the low variation of performances of a model is always desirable.

$\bullet$ Lines $334$ to $337$, page $9$ (Section $4$): the advantage of BR-MDP over DR-MDP in Table $1$ is too minor (59.64(1.42) versus 60.00(0.00)). Hence this may not be an ``evident illustration".

$\bullet$ Line $342$, page $9$ (Section $4$): to better illustrate the claim that "the difference is small", I would suggest the authors provide a table to report the relative gaps (in \%) between $\tilde{V}^*_0(s_0,\mu_0)$ and $V^*_0(s_0,\mu_0)$.

$\bullet$ Could the authors briefly clarify why the exact solution of CVaR BR-MDP is computationally available in the experiments (Since they state in the first paragraph of Section $3$ that this exact solution is hard to obtain)?

$\bullet$ The sizes of the historical data are fixed to be $10$ in both experiments. I would suggest the authors vary the sample sizes to better examine the performances of the models.

$\bullet$ Could the authors explain why the computation times of "Nominal" and "DR-MDP" are not reported in Table $1$?

$\bullet$ I would suggest the authors explain why the performances of DR-MDP are not reported in Figure $1{\rm a}$.

$\bullet$ Line $121$, page $3$ (Section $2$): "$Z$" should be "$X$"?

$\bullet$ Lines $169$ to $170$, page $4$ (Section $2$): the right-hand side of the inequality, should it be $\mathbb{E}_{\xi_1}$ rather than $\mathbb{E}_{\xi_1\vert\xi_0}$ (based on the independence of $\xi_t$'s described in Lines $135$ to $136$)?

$\bullet$ Lines $205$ to $206$, page $5$ (Section $2$): should it be "...$V_{t+1}^*(s_{t+1},\mu_{t+1})...$"?

$\bullet$ Lines $211$ to $213$, page $6$ (Section $3$): the statement should be: "$V_t^*=...$, where $\alpha_t(s_t,\theta)=...$". Also, what is the range of $t$ for the equations to hold?

$\bullet$ Lines $234$ to $235$, page $6$ (Section $3$): $f(\xi\vert\theta)$ should be $f(\xi;\theta)$ for consistency.

$\bullet$ Algorithm $2$, page $7$: the algorithm should have only one "\textbf{output}".

$\bullet$ Table $1$, page $9$ (Section $4$): the rows of the two tables should be aligned correspondingly.

$\bullet$ Figure $1$, page $9$ (Section $4$): it might be better to divide Figure $1$ into two figures.



**Limitations:**

Yes

**Strengths And Weaknesses:**

Strengths:
Overall, the paper is well-written. I especially appreciate the design of the approximate dynamic programming to overcome computational inefficiency.

Weaknesses:
1. Please be very specific if the proposed framework only works with one-dimensional uncertainty, i.e., $\theta$. If yes, please acknowledge this well and provide strong justifications.
2. I think several key assumptions for the whole framework should be listed more clearly and frankly. For example, when talking about the $\alpha$-function representation (a major focus of this paper), the authors hide the critical assumption that "the parameter space is a finite set" quite "deep", which does not seem to be proper to me. For another example, for Theorem 3.2, the authors require joint convexity while not providing any justification (e.g., is it easy to satisfy).
3. I would encourage the authors to talk more about how they are motivated by "POMDP" and discuss the connections in more detail.
4. In the experiments, the results are not convincing enough (\textit{e.g.}, only one sample size is considered, and the advantage of the proposed model is not obvious,

---

> ### Author Response · Authors · 2022-08-02
> **Response to Reviewer sCox I**
>
> Thank you very much for the valuable time you have spent reviewing our work. Below are our responses to the comments and questions you have.
>
> - **Please be very specific if the proposed framework only works with one-dimensional uncertainty, i.e., $\theta$. If yes, please acknowledge this well and provide strong justifications**.
>
> Thank you for your suggestion. The proposed formulation and algorithm work for multi-dimensional parameter space. We have added the following descriptions in Section 3 of the revised version to make it clear: $\theta \in \Theta \subset \mathbb{R}^d$, where $d$ is the dimension of the parameter $\theta$.
>
> - **I think several key assumptions for the whole framework should be listed more clearly and frankly. For example, when talking about the alpha-function representation (a major focus of this paper), the authors hide the critical assumption that "the parameter space is a finite set" quite "deep", which does not seem to be proper to me. For another example, for Theorem 4.3, the authors require joint convexity while not providing any justification (e.g., is it easy to satisfy)**.
>
> Thank you for your valuable suggestion. To generalize our proposed algorithm to a continuous parameter space is possible, where one can replace the summation over $\theta$ by integral. It does not pose a challenge to the technical proof, but it will bring computational difficulty in Bayesian updating of the posterior distribution since Bayesian updating on a general space often does not admit closed form. To overcome this computational difficulty, we can impose a conjugate assumption on the prior distribution and likelihood function such that the posterior distribution has closed form and can be computed easily. We have lifted the finite assumption on the parameter space and have added more discussion on the Bayesian updating. As for the joint convexity, we have added the following remarks following Theorem 4.3: ``
>
> The jointly convex assumption in Theorem 4.3 is common for gradient-based algorithms for solving multi-stage decision making problems (e.g. [1]). It is satisfied in many real-world problems such as inventory control (e.g. [2]) and portfolio optimization (e.g. [3]) etc.''.
>
> We have been more clear when stating the key assumptions in the revised version.
>
> - **I would encourage the authors to talk more about how they are motivated by ``POMDP'' and discuss the connections in more detail**.
>
> Thank you for your valuable suggestion. The connection between BR-MDP and POMDP is motivated by the fact that the posterior distribution in BR-MDP is exactly like the belief state (which is the posterior distribution of the unobserved state given the history of observations) in POMDPs. However, after we did the work, we found [4] in the literature also reformulated a Bayes-adaptive MDP into a POMDP. We have discussed the connections in more details in the revised version.
>
> - **The results are not convincing enough (e.g., only one sample size is considered, and the advantage of the proposed model is not obvious**.
>
> - **The sizes of the historical data are fixed to be 10 in both experiments. I would suggest the authors vary the sample sizes to better examine the performances of the models**.
>
> Thank you for your comment. Following your suggestion, we include more results by varying the data size $N=5,10,100$ in the gambler's betting problem, shown in Table 2 in the revised version. In particular, when the data size is very small ($N=5$), our proposed BR-MDP formulation provides more robustness than the nominal approach by having a much smaller standard deviation. It would be tempting to associate the good performance of our proposed formulation to much lower average cost compared to the nominal approach. However, this is an inappropriate interpretation. Note that our proposed BR-MDP framework tries to avoid a scenario where a solution performs well under the estimated model but performs badly under the true model, by possibly giving up some good expected performance and trading for more confidence about the actual performance of a solution. When the data size is very small, the MLE estimator in the nominal approach varies in each replication, which leads to large variance of the obtained policy over different replications. Instead, our BR-MDP formulation mitigates this parameter uncertainty (or model uncertainty, epistemic uncertainty etc.) and produces policy that is stable over different replications. Also note that DR-MDP is the most conservative formulation: for almost every dataset it is given, it produces the policy ``not to bet''. This conservativeness is not desirable if the true winning rate $\theta^c>0.5$. In summary, the numerical results demonstrate that our BR-MDP formulation seeks a trade-off between the expected performance and the robustness in the actual performance.

---

> ### Author Response · Authors · 2022-08-02
> **Response to Reviewer sCox II**
>
> - **What is the meaning of $\Xi_t$. Is it a random variable or a set**.
>
> Thank you for your question. We have condensed our notations and got rid of $\Xi_t$ in the revised version. We meant by $\Xi_t$ the set of randomness realizations that satisfy the state transition $s_{t+1}=g_{t}\left(s_{t}, a_{t}, \xi_{t}\right)$. Please see the following statement in the revised version: "The state equation together with the distribution of $\xi_t$ uniquely determines the transition probability of the MDP, i.e., $\mathcal{P}(s_{t+1}\in S'|s_t,a_t)=\mathbb{P}(\{\xi_t \in \Xi: g_t(s_t,a_t,\xi_t)\in S'\}|s_t,a_t)$, where $S'$ is a measurable set in $\mathcal{S}$."
>
> - **Line 164, page 4 (Section 2): by ``To illustrate'', I expect the authors to demonstrate the time consistency of the BR-MDP, but they just show that the objective value of BR-MDP serves as an upper bound for the objective of the one considering static risk functional. I would suggest the authors relate this to the concept of time consistency**.
>
> Thank you for your valuable suggestion. The upper bound is used to show that the static risk functional always yields a higher total expected cost than the nested risk functional, illustrating the benefit of nested risk functional which originates from time consistency. But since it is confusing and also due to page limit, we have moved this to the Appendix and rewritten this part.
>
> - **Based on formula (2), should it be $\mathcal{C}_1(s_1)$ instead of $\mathcal{C}_1(s_1,a_1,\xi_1)$**.
>
> Thank you for pointing it out. Yes, you are right. The argument can be made by including one more stage, such that the last stage is $\mathcal{C}_2(s_2)$.
>
> - **The definition of $\Gamma_t$ is confusing. Does it imply that the cardinalities of $\Gamma_0, \Gamma_{1}, \cdots$ are all $|\mathcal{A}|$**.
>
> Thank you for pointing it out. Yes, you are right that the cardinalities of $\Gamma_0, \Gamma_{1}, \cdots$ are all $|\mathcal{A}|$. We realize the confusion may arise when we show the difficulty of computing the set $\Gamma_t$. Suppose we are given the set $\Gamma_{t+1}$ and we want to compute $\Gamma_{t}$. From the alpha-function representation in Proposition 4.1, we need to first determine the minimizer $\alpha_{t+1}^{\*(\xi)}$, which attains the minimum of $\int_{\Theta} \alpha_{t+1}(s_{t+1}, \theta) \frac{\mu_t(\theta)f(\xi;\theta)}{\int_{\Theta}\mu_t(\theta)f(\xi;\theta)d\theta}d\theta$. We use superscript $\xi$ to explicitly show that for each $\xi$, we need to find a minimizer $\alpha_{t+1}^{\*(\xi)}$ in set $\Gamma_{t+1}$. Since for each $\xi$, there are $|\Gamma_{t+1}|$ candidates for $\alpha_{t+1}^{\*(\xi)}$ (which means one has to search over those candidates to find the minimizer), there are a total of $|\Gamma_{t+1}|^{|\Xi|}$ candidates for $\int_{\Xi}f(\xi;\theta)\left(C_t(s_{t},a_{t},\xi)+ \min_{\alpha_{t+1}} \int_{\Theta} \alpha_{t+1}(s_{t+1}, \theta) \frac{\mu_t(\theta)f(\xi;\theta)}{\int_{\Theta}\mu_t(\theta)f(\xi;\theta)d\theta}d\theta \right)d\xi,$ and thus a total of $|\mathcal{A}||\Gamma_{t+1}|^{|\Xi|}$ candidates for $\Gamma_t$. Please see the revised version for more clear presentation.
>
> - **What is the value of $M_C$**.
>
> Thank you for your question. $M_C$ is the maximum customer demand, which is explained right after the notation.
>
> [1] Guigues, V., Shapiro, A. and Cheng, Y., 2021. Risk-averse Stochastic Optimal Control: an Efficiently Computable Statistical Upper Bound. arXiv preprint arXiv:2112.09757.
>
> [2] Chang, H.S., Fu, M.C., Hu, J. and Marcus, S.I., 2005. An Adaptive Sampling Algorithm for Solving Markov Decision Processes. Operations Research, 53(1), pp.126-139.
>
> [3] Hazan, E., 2016. Introduction to Online Convex Optimization. Foundations and Trends in Optimization, 2(3-4), pp.157-325.

---

> ### Author Response · Authors · 2022-08-02
> **Response to Reviewer sCox III**
>
> - **It seems to me that the low variation of performances of a model is always desirable**.
>
> Thank you for your comment. A low variation of performance of a model is not always desirable if it sacrifices too much mean performance. In general, a robust approach strives for a good balance between the variation and the average of performance. For example, consider the following betting example: the true winning rate $\theta^c=0.8$. If the gambler always chooses not to bet at any time stage (for example with distributionally robust formulation), the variation of the performance of this model is always 0. However, This model is not desirable as the gambler has a high winning rate and should choose to bet. As another example, consider a newsvendor problem. The newsvendor can choose nothing to sell, which means the cost is always 0. However, she may also have a high chance of making money if she chooses to order some newspaper to sell, when the distribution of the customer demand has a small probability mass on zero. As a result, the low variation of performances is not always desirable, and it is more important to balance the variation and mean performance.
>
> - **The advantage of BR-MDP over DR-MDP in Table 1 is too minor (59.64(1.42) versus 60.00(0.00)). Hence this may not be an ``evident illustration"**.
>
> Thank you for your comment. We found a mistake in our previous implementation in the betting problem: we added a constant $c$ (which is an upper bound on stage-wise cost) to the cost at each time stage to ensure the stage-wise cost non-negative (as required by our algorithm), but forgot to subtract $cT$ (which is the total added constant over $T$ horizons) from the algorithm output to recover the total cost of the problem. As a result, the added constant $cT$ dominates the total cost and obscures the difference between different formulations. After correcting this mistake, we can see more difference between BR-MDP and DR-MDP (see Table 1): for example, in the betting problem with true parameter $\theta^c=0.55$, BR-MDP (exact, $\alpha=0.4$) yields mean cost  -2.28 and variance 4.28, while DR-MDP yields mean cost = 0.00, and variance = 0.00. In the inventory control problem, the advantage of BR-MDP (mean 81.63, variance 2.27) over DR-MDP (mean 99.77, variance 0.00) is also significant.
>
> - **To better illustrate the claim that ``the difference is small'', I would suggest the authors provide a table to report the relative gaps**.
>
> Thank you for your valuable suggestion. We have reported the relative gap in the appendix in our revised version.
>
> - **Could the authors briefly clarify why the exact solution of CVaR BR-MDP is computationally available in the experiments (Since they state in the first paragraph of Section 4 that this exact solution is hard to obtain)**.
>
> Thank you for your question. To obtain the "exact" (more precisely, should be close-to-exact) BR-MDP optimal value function, we discretize the continuous state, i.e., the posterior distribution, with small step size 0.1, which results in very large state space, and then we conduct dynamic programming on the discretized problem to obtain the optimal value function. This is a brute-force way to compute the "exact" value function, and that's why the computational time for the exact BR-MDP formulation is extremely large compared to the approximate formulation. We have clarified this in the revised version.
>
> - **Could the authors explain why the computation times of "Nominal" and "DR-MDP" are not reported in Table 1**.
>
> Thank you for your question. In the revised version, we include the computational times of nominal and DR-MDP in Table 1.
>
> - **I would suggest the authors explain why the performances of DR-MDP are not reported in Figure 1a**.
>
> Thank you for your suggestion. In the revised version, we have included the performance of DR-MDP in Figure 1a, which is a vertical line at 60.
>
> - **Possible typos: Line 121: Z should be X; Line 169: should it be $E_{\xi_1}$; Line 205: should it be $V_{t+1}^{\*}(s_{t+1},\mu_{t+1})$; Line 211: range of t should be $0,1,\cdots,T-1$; Line 234: $f(\xi;\theta)$ instead of $f(\xi|\theta)$ for consistency; Algorithm 2: the algorithm should have only one output; Table 1: the rows of the two tables should be aligned correspondingly**.
>
> Thank you for catching the typos. Thank you for your corrections. We have corrected them in the revised version.
>
> [4] Poupart, P., Vlassis, N., Hoey, J. and Regan, K., 2006. An Analytic Solution to Discrete Bayesian Reinforcement Learning. In Proceedings of the 23rd international conference on Machine learning, pp. 697-704.
>
> [5] Shapiro, A., 2021. Tutorial on Risk Neutral, Distributionally Robust and Risk Averse Multistage Stochastic Programming. European Journal of Operational Research, 288(1), pp.1-13.

---

### Official Review · Reviewer_YVjo · 2022-07-09

**Rating:** 4
**Confidence:** 3
**Soundness:** 3 good
**Presentation:** 3 good
**Contribution:** 2 fair

**Summary:**

The paper presents a Bayesian risk approach for finite-horizon MDPs under uncertainty. By using some techniques from Bayesian risk optimization, the authors develop an approximation algorithm to solve the MDP problem. The algorithm is then supported by numerical experiments based on two finite-horizon offline planning problems.  The paper is well-written. Both robust MDP and Bayesian risk optimization are important areas in Machine Learning/Optimization. Thus, the problem considered is interesting and worth investigating.

**Questions:**

- Page 4, line 2 is unclear, what is $\Xi_t$. It was not defined before
- The pdf and the version in the supplement are different. Any explanation for this?
- Why is the conclusion missing?
- In (6), both $V_{1}$  and $V_{t+1}$ on the both sides depend on $\mu_t$. Should it be $\mu_{t+1}$ on the right hand side?

**Ethics Review Area:**

["I don’t know"]

**Limitations:**

There would be no potential negative societal impact of this work.

**Strengths And Weaknesses:**

## Strengths:
The formulation is new.  The paper is technically sound and well written/organized.
The combination of MDP under uncertainty and Bayesian Risk Optimization seems interesting and promising.

## Weaknesses:
 I believe the paper is not ready for publication due to several limitations as stated below:

Robust MDP and Bayesian risk optimization formulations are widely studied and well understood. Thus, the results developed in the paper are quite **trivial**, given all the techniques we have in the literature. For example, Section 2 (including Algorithm1) only presents already known or trivial material. The alpha-function representation is also a direct result of equation (5) and some techniques in the Bayesian risk optimization literature. The Bellman equation in (6) is just a standard Bellman equation with a continuous state space, for which several efficient (approximation) algorithms exist, but I do not see an explicit comparison of the proposed algorithm against existing ones.

The paper only focuses on finite-horizon MPDs which makes it limited, as many applications will require infinite MDPs

Some assumptions are vague and require explanations, for example, the authors assume on Page 5 that \Theta is finite. It is true that we can always discretize a continuous space, but it raises several questions: How to discretize, how many discrete samples would be needed?  approximation errors as a function of the discretization points...

The authors claim that their algorithm can be extended easily to other coherent risk measures. To support this point, it is better to provide formulations and results with other risk measures.

---

> ### Author Response · Authors · 2022-08-02
> **Response to Reviewer YVjo I**
>
> Thank you very much for the valuable time you have spent reviewing our work. Below are our responses to the comments and questions you have.
>
> - **Robust MDP and Bayesian risk optimization formulations are widely studied and well understood. Thus, the results developed in the paper are quite trivial, given all the techniques we have in the literature**.
>
> Thank you for your comment. With all due respect, we argue that Bayesian risk optimization has not been widely studied and has only studied in the limited setting of static (i.e., one-stage) optimization. In fact, Bayesian risk optimization, abbreviated as BRO, was proposed very recently and studied in only a handful of literature ([1], [2], [3]). BRO is a novel framework to replace the widely-used distributionally robust optimization (DRO) framework in static optimization, and has never been explored for multi-stage and dynamic settings. To the best of our knowledge, we are the first to extend the single-stage BRO formulation to the multistage setting, especially in the context of Markov decision process, optimal control, or reinforcement learning. While robust MDP is widely studied, we deviate from the mainstream literature of robust MDP in terms of formulation and the consequent solution method. The starting point of our paper is  discussing the limitations of the mainstream robust MDP formulations (such as conservativeness and lack of time consistency; see the second paragraph of the introduction), and therefore, we propose a new formulation. This new formulation, which has a nested structure and uses an Bayesian approach, requires significantly different solution approaches from solving mainstream robust MDPs. Roughly speaking, mainstream robust MDPs require solving a mini-max problem in the Bellman equation, while our formulation only needs to solve a minimization problem but introduces an additional belief state that needs to be (Bayesian) updated over time stages. Therefore, the results in this paper, from formulation to solution methods, are not trivial at all.
>
> - **For example, Section 2(including Algorithm 1) only presents already known or trivial material**.
>
> Thank you for your comment. Section 2 (now Section 3 in the revised version) presents some already known material to pave the way for introducing our new formulation, while the rest of Section 2 presents new and non-trivial materials. Specifically, Section 2 (Preliminaries and Problem Formulation) presents known results (preliminaries) in Sec. 2.1 to introduce necessary background for this paper and in part of Sec. 2.2 to introduce the definition of an MDP and its parameter uncertainty. We then propose a new formulation in the second half of Sec. 2.2 and show its Bellman optimality and time consistency in Sec. 2.3. Please note this formulation and result are not trivial, because the time consistency (and hence the existence of the Bellman optimality) is exactly what our new formulation differs from the mainstream robust MDPs. Algorithm 1 is a natural consequence of the Bellman equation, which is listed there as the "idealized" benchmark (as opposed to the "practical" solution developed in the following section). We have tried to clarify in the revised version.
>
> [1] Zhou, E. and Xie, W., 2015. Simulation Optimization When Facing Input Uncertainty. In 2015 Winter Simulation Conference, pp. 3714-3724.
>
> [2] Wu, D., Zhu, H. and Zhou, E., 2018. A Bayesian Risk Approach to Data-driven Stochastic Optimization: Formulations and asymptotics. SIAM Journal on Optimization, 28(2), pp.1588-1612.
>
> [3] Cakmak, S., Wu, D. and Zhou, E., 2021. Solving Bayesian Risk Optimization via Nested Stochastic Gradient Estimation. IISE Transactions, 53(10), pp.1081-1093.

---

> > ### Comment · Reviewer_YVjo · 2022-08-08
> > **Thanks for the responses**
> >
> > I thank the authors for the responses, especially their effort to explain the novelty of the work and how the model differs from existing works. I agree that the optimization models and methods are not as trivial as I thought. However, given all the existing works in the literature, I believe that the use of $\alpha$ function and the development of the approximation procedure in the context is quite straightforward. Some issues still remain, e.g., finite-horizon MDP. I have raised my scores accordingly, but I still think that the paper is not ready for publication.

---

> > > ### Author Response · Authors · 2022-08-09
> > > **Follow-up on thanks for the response**
> > >
> > > We thank you for acknowledging our work and effort! We also appreciate that you raised your score.
> > >
> > > We would like to emphasize again that the POMDP literature only considers alpha-function representation for the risk neutral case and does not generalize to the risk-averse setting. More specifically, the risk-neutral alpha-function representation makes use of the piecewise linearity of the optimal value function, which does not hold in the risk-averse setting. Moreover, the CVaR risk functional adds additional complexity to the alpha function approximation due to the optimization over $u_t$.
> > >
> > > With all due respect, we do not think the restriction to finite-horizon MDPs is an issue for a paper (a general paper, not just our paper). As you probably know, the treatment of infinite-horizon MDPs is often different from finite-horizon MDPs. More specifically, an infinite-horizon MDP is essentially a fixed-point problem, which entails several classes of methods including value iteration, policy iteration, and linear programming methods. None of these methods are used for finite-horizon MDPs, although value iteration (for infinite horizon) is similar to dynamic programming (for finite horizon) in format. Classical books on MDPs, such as "Dynamic Programming and Optimal Control" by Dimitri Bertsekas, also treat finite-horizon and infinite-horizon MDPs in separate chapters. Infinite-horizon MDP is definitely an important class of problems, but there is nothing wrong for a paper to focus on finite-horizon MDPs, which are important problems as well and have wide applications.

---

> ### Author Response · Authors · 2022-08-02
> **Response to Reviewer YVjo II**
>
> - **The alpha-function representation is a direct result of equation (5) and some techniques in the Bayesian risk optimization literature**.
>
> Thank you for your comment. With all due respect, we disagree with the reviewer's comment. First, the alpha-function representation has nothing to do with Bayesian risk optimization (BRO is only used in the problem formulation of BR-MDP and has nothing to do with the solution methods to BR-MDP). Second, we showed the exact alpha-function representation (in Sec. 3.1) and an approximate representation (in Sec. 3.2), both of which are not direct result or trivial extension of previous results. The exact alpha-function representation differs from the POMDP literature which only consider alpha-function representation for the risk neutral case  ([4], [5]) and do not generalize to the risk-averse setting. More specifically, the risk-neutral alpha-function representation make use of the piecewise linearity of the optimal value function, which does not hold in the risk-averse setting. Moreover, the exact alpha-function, even in the risk-neutral case,  suffers from the ``curse of time'' in the sense that the number of alpha functions grows exponentially over time. In the risk-averse setting, this difficulty is even more severe because the CVaR risk functional adds additional complexity to the alpha function approximation due to the optimization over $u_t$. To overcome this difficulty, we further develop an approximate alpha-function representation that keeps a constant small number of alpha-functions over time for a fixed $u$, combined with a gradient descent algorithm that optimizes over $u$ in the outer loop.
>
> - **The Bellman equation in (6) is just a standard Bellman equation with a continuous state space, for which several efficient (approximation) algorithms exist,but I do not see an explicit comparison of the proposed algorithm against existing ones**.
>
> Thank you for your comment. The Bellman equation  in equation (6) differs from the standard Bellman equation of MDPs in two aspects. First, we introduce the posterior distribution as an additional continuous state, while there is no such state in the standard Bellman equation of MDPs. While  the posterior distribution can be theoretically regarded as just another continuous state, it is often infinite dimensional (for general distributions) or lives in a multi-dimensional simplex (for discrete distributions) and hence creates unique difficulty and opportunity for computation (e.g. the alpha-function representation we explored is unique for this type of continuous-state MDP but does not hold for general continuous-state MDPs). Second, we impose the risk functional CVaR to quantify the uncertainty brought by the unknown parameter and CVaR introduces an additional variable $u$ and adds another layer of optimization in the Bellman equation, while there is no quantification of parameter uncertainty in standard Bellman equations for MDPs.
>
> As explained above, our Bellman equation is different from the usual Bellman equation, and hence the standard methods for solving continuous-state MDPs do not generalize easily to our problem. With that said, we did found some efficient approximation algorithms for such (continuous-state, risk-averse for parameter uncertainty) Bellman equations, and probably the closest work to ours in the literature is [7]. In that work, the authors proposes to solve a CVaR risk functional over the **total cost** and simultaneously address both epistemic and aleatoric uncertainty. However, their formulation, with a static risk measure, will lead to a time-inconsistent behavior, where the optimal policy at the current time stage can become suboptimal in the next time stage simply because a new piece of information is revealed (see [8]). This may not be problematic as their consider an **online** RL setting, where after making a decision, the agent can interact with the true environment, receive the corresponding reward and take the state transition. Whereas in this work, we consider an **offline** setting, where there is no interaction with the true environment when we make the decision. Directly comparing with their approach in our **offline** setting may not be a fair comparison. On the other hand, since we motivate the problem formulation from the conservative distributionally robust MDP (DR-MDP) formulation, we did include comparison with the DR-MDP in our numerical results.

---

> ### Author Response · Authors · 2022-08-02
> **Response to Reviewer YVjo III**
>
> - **The paper only focuses on finite-horizon MDPs which makes it limited, as many applications will require infinite MDPs**.
>
> Thank you for your comment. We admit that the algorithm proposed in this work only focuses on solving a finite-horizon MDPs. We do have some initial theoretical results for infinite-horizon MDPs (not included in this paper), but the development of algorithms will require very different machinery (completely different from the alpha-function representation used in the finite-horizon case) and probably warrants a separate paper. Therefore, we will leave the theoretical analysis and algorithm development for infinite-horizon case to a future work.
>
> - **Some assumptions are vague and require explanations, for example, the authors assume on Page 5 that $\Theta$ is finite**.
>
> Thank you for pointing it out. To generalize our proposed algorithm to a continuous parameter space is possible, where one can replace the summation over $\theta$ by integral. It does not pose a challenge to the technical proof, but it will bring computational difficulty in Bayesian updating of the posterior distribution since Bayesian updating on a general space often does not admit closed form. To overcome this computational difficulty, we can impose a conjugate assumption on the prior distribution and likelihood function such that the posterior distribution has closed form and can be computed easily. We have lifted the finite assumption on the parameter space and have added more discussion on the Bayesian updating.
>
> - **The authors claim that their algorithm can be extended easily to other coherent risk measures. To support this point, it is better to provide formulations and results with other risk measures**.
>
> Thank you for your valuable suggestion. Consider coherent risk measure based on KL divergence, as an example given in [9]. Here the risk functional takes the form $R_{\epsilon}(Z)=\inf_{\gamma, \lambda>0}\{\lambda \epsilon+\gamma+\lambda e^{-\gamma / \lambda} \mathbb{E}e^{Z / \lambda}-\lambda\}$, where $\epsilon$ is the user-defined ambiguity set size. Given $\lambda>0$, the minimizer $\gamma=\lambda \ln \mathbb{E}\left[e^{Z / \lambda}\right]$. We can apply the same technique to approximate the alpha functions for a given vector $\lambda_0,\cdots,\lambda_{T-1}$, and then apply gradient descent on the approximate value function. Due to the page limit and the main focus on the CVaR risk measure, we have included more details in the revised supplementary material.
>
> - **Page 4, line 2 is unclear, what is $\Xi_t$. It was not defined before**.
>
> Thank you for your question. We have condensed our notations and got rid of $\Xi_t$ in the revised version. We meant by $\Xi_t$ the set of randomness realizations that satisfy the state transition $s_{t+1}=g_{t}\left(s_{t}, a_{t}, \xi_{t}\right)$. Please see the following statement in the revised version:  "The state equation together with the distribution of $\xi_t$ uniquely determines the transition probability of the MDP, i.e., $\mathcal{P}(s_{t+1}\in S'|s_t,a_t)=\mathbb{P}(\{\xi_t \in \Xi: g_t(s_t,a_t,\xi_t)\in S'\}|s_t,a_t)$, where $S'$ is a measurable set in $\mathcal{S}$.''
>
> - **The pdf and the version in the supplement are different. Any explanation for this**.
>
> Thank you for pointing it out. The pdf was part of the supplementary material. In the revised supplementary material, we only include the appendix.
>
> - **Why is the conclusion missing**.
>
> Due to the page limit, we intentionally left out the conclusion section in our current version. But since you think that is important, we have included the conclusion section in the revised version by shortening the introduction and leaving room for the conclusion.
>
> - **Possible typos**.
>
> Thank you for catching the typos. Thank you for your correction. Yes, the RHS of equation (6) should be $V^*_{t+1}(s_{t+1},\mu_{t+1})$. It is a typo and we have fixed it in the revised version.
>
> [4] Smallwood, R.D. and Sondik, E.J., 1973. The Optimal Control of Partially Observable Markov Processes over a Finite Horizon. Operations research, 21(5), pp.1071-1088.
>
> [5] Poupart, P., Vlassis, N., Hoey, J. and Regan, K., 2006. An Analytic Solution to Discrete Bayesian Reinforcement Learning. In Proceedings of the 23rd international conference on Machine learning, pp. 697-704.
>
> [6] Zhou, E., 2012. Optimal Stopping under Partial Observation: Near-value Iteration. IEEE Transactions on Automatic Control, 58(2), pp.500-506.
>
> [7] Rigter, M., Lacerda, B. and Hawes, N., 2021. Risk-averse Bayes-adaptive Reinforcement Learning. Advances in Neural Information Processing Systems, 34, pp.1142-1154.
>
> [8] Shapiro, A., 2021. Tutorial on Risk Neutral, Distributionally Robust and Risk Averse Multistage Stochastic Programming. European Journal of Operational Research, 288(1), pp.1-13.
>
> [9] Guigues, V., Shapiro, A. and Cheng, Y., 2021. Risk-averse Stochastic Optimal Control: an Efficiently Computable Statistical Upper Bound. arXiv preprint arXiv:2112.09757.

---

### Official Review · Reviewer_wNPu · 2022-07-11

**Rating:** 5
**Confidence:** 3
**Soundness:** 2 fair
**Presentation:** 1 poor
**Contribution:** 2 fair

**Summary:**

This paper provides a Bayesian risk Markov Decision Process (BR-MDP) formulation to address parameter uncertainty in MDP.
- The paper uses a Bayesian posterior distribution as opposed to ambiguity set, and imposes a risk functional on the objective function with respect to the posterior distribution.
- Specifically, the authors assume the distribution of the randomness in the system belongs to some parametric family, and model the uncertainty over the unknown parameters via Bayesian posterior distributions.
- Furthermore, a CVaR risk functional taken with respect to the posterior distribution on the expected total cost is proposed in a nested form to promote time-consistency solutions.
- Finally, the authors derived a dynamic programming solution with an augmented state that incorporates the posterior information and proposes an analytical approximate solution to BR-MDP.

**Questions:**

- Section 3.2, The first paragraph is hard to follow, the rewriting of equation 6 is not clear.
- Line 233, it says $V_t(s_t, \mu_t)$  stands for the "optimal" value function, what does  $V^*_t(s_t, \mu_t)$ (equation 6) stand for?
- Does the proposed method scale when the parameter space is high-dimensional?


Possible typos:
- Equation 6: rhs, $V_{t+1}^*(s_{t+1}, \mu_t)$ should be $V_{t+1}^*(s_{t+1}, \mu_{t+1})$
- Line 234-235, both $f(\epsilon; \theta)$ and $f(\epsilon|\theta)$ appeared in this equation. Do they mean the same thing?

**Limitations:**

The authors did not addressed any limitations and potential negative societal impact of their work.


## Suggestions:
###  Several ways to improve the clarity of the paper:
  - Condense introduction part (the current version may well suited for a journal paper; but for conference paper, the space of the introduction is a bit too much)
  - Check the notations and remove redundant information. e.g. equation 4 appears in multiple places (line 177, line 213 (integral is replaced by \Sum).





**Strengths And Weaknesses:**

## Strength
- The idea of using Bayesian posterior to model the uncertainty in the parameters of MDP is intuitive and will improve the overly conservative problem in distributionally robust-MDP method.
- Compressive theoretical analysis of the approximation are provided.
- The proposed approximation method reduce the computation time significantly while not influencing the performance.


## Weakness:
- The readability of the paper suffers from inconsistency and heavy notations.
- The paper seems unfinished since it lacks a conclusion section.
- The effectiveness of the paper is only evaluated on simple synthetic experiments.

---

> ### Author Response · Authors · 2022-08-02
> **Response to Reviewer wNPu I**
>
> Thank you very much for the valuable time you have spent reviewing our work. Below are our responses to the comments and questions you have.
> - **Condense introduction part (the current version may well suited for a journal paper; but for conference paper, the space of the introduction is a bit too much)**.
> - **The readability of the paper suffers from inconsistency and heavy notations**.
> - **Check the notations and remove redundant information. e.g. equation 4 appears in multiple places (line 177, line 213 (integral is replaced by sum)**.
>
> Thank you for your comments and suggestions. The majority of the introduction focuses on motivating our problem formulation and has a long explanation of why we consider the Bayesian risk optimization with a time-consistent multistage formulation. We have moved parts of the comparison with existing literature on robust optimization after the introduction to keep it short, while maintaining a motivating and clear introduction.
>
> The heavy notations are due to the complexity of the considered problem (risk-averse settings, CVaR, alpha function approximation etc.).  In our revised version we have tried to condense and streamline the notations as much as possible, and also provided a table summarizing main notations in the appendix for future reference.
>
> - **The paper seems unfinished since it lacks a conclusion section**.
>
> Thank you for your comment. Due to the page limit, we intentionally left out the conclusion section in our current version. But since you think that is important, we have included the conclusion section in the revised version by shortening the introduction and thus leaving room for the conclusion.
>
> - **The effectiveness of the paper is only evaluated on simple synthetic experiments**.
>
> Thank you for your comment. Applying the proposed algorithms to more complicated domains with real-world data may be left as a future work. The focus of this work is on bringing out the novel problem formulation and designing a new alpha-function approximation approach to solve it efficiently. The two numerical examples we use come from [1] and [2], which were respectively published in NeurIPS 2021 and in a leading journal *Operations Research*. These examples allow  better understanding of our results by comparing to the true optimal solutions (which can be obtained thanks to simple structure of these problems), to the empirical approach (from the frequentist perspective), and to the distributionally robust approach in the offline setting. From these comparisons, our method shows less conservativeness and demonstrates the benefit of considering future data realization in our proposed formulation.
>
> - **Section 4.2, The first paragraph is hard to follow, the rewriting of equation (6) is not clear**.
>
> Thank you for your comment. Equation (6) defines the optimal value function with the augmented state $(s,\mu)$. It essentially solves a minimization problem, where the minimization is taken with respect to action $a$ and variable $u$ used in the CVaR expression. Then we introduce the optimal $Q$ function, which is a function of augmented state $(s,\mu)$ and also $a$ and $u$. The optimal value function is equivalent to the minimum of the optimal $Q$ function. This completely parallels the optimal value function and $Q$ function in the traditional MDP and RL literature.  We have made it more clear in the revised version.
>
> - **Line 233, it says $V_t(s_t,\mu_t)$ stands for the ``optimal'' value function, what does $V_t^{*}(s_t,\mu_t)$ (equation (6)) stand for**.
>
> Thank you for raising this question. Note that the true optimal value function $V^{*}_t(s_t,\mu_t)$ defined in equation (6) is solved by minimizing over $a_t$ and $u_t$. The ``optimal'' value function $V_t(s_t,\mu_t)$ is solved by minimizing over $a_t$ but with $u_t$ fixed. The idea is that we first conduct alpha function approximation with a fixed $u$ vector. In that case we can control the number of alpha functions. We have rewritten to make it clear in the revised version.
>
> [1] Rigter, M., Lacerda, B. and Hawes, N., 2021. Risk-averse Bayes-adaptive Reinforcement Learning. Advances in Neural Information Processing Systems, 34, pp.1142-1154.
>
> [2] Chang, H.S., Fu, M.C., Hu, J. and Marcus, S.I., 2005. An Adaptive Sampling Algorithm for Solving Markov Decision Processes. Operations Research, 53(1), pp.126-139.

---

> ### Author Response · Authors · 2022-08-02
> **Response to Reviewer wNPu II**
>
> - **Does the proposed method scale when the parameter space is high-dimensional**.
>
> Thank you for your question. First, the proposed formulation and algorithm work for a multi-dimensional parameter space. We will add the following descriptions in Section 3 of the revised version to make it clear: $\theta \in \Theta \subset \mathbb{R}^d$, where $d$ is the dimension of the parameter $\theta$. Second, the proposed algorithm can scale easily to a high-dimensional parameter space. A high-dimensional parameter space leads to high-dimensional integration in both Bayesian updating and dynamic programming. For high-dimensional integration, one can resort to Monte Carlo integration which enjoys a convergence rate of $1/\sqrt{\text{sample size}}$ and in independent of the dimension.
>
> - **Possible typos**.
>
> Thank you for catching the typos. Yes, the RHS of equation (6) should be $V^*_{t+1}(s_{t+1},\mu_{t+1})$; $f(\xi;\theta)$ and $f(\xi|\theta)$ mean the same thing. We have corrected them in the revised version.
>
> - **The authors did not addressed any limitations and potential negative societal impact of their work**.
>
> Thank you for your comment. We report no negative societal impacts, including but not limited to potential malicious or unintended uses, environmental impact, fairness considerations, privacy considerations, or security considerations. Combined with your previous suggestion of adding the conclusion section, we have discussed the limitation of the work and future direction in the conclusion section in our revised version.

---

### Author Response · Authors · 2022-08-08
**A gentle reminder to reviewers**

Thank you for your time reviewing our paper. We have done our best to address your concerns, summarized the rationale for doing so, and modified the paper accordingly. We just uploaded a new version that highlights the main changes in red. If you have any further questions, please feel free to leave us a message and we are happy to discuss it. Please note the discussion period closes soon, so we will greatly appreciate your feedback either in discussions or scores.

---

### Meta-Review · Area_Chair_PxWY · 2022-08-27

**Recommendation:** Accept
**Confidence:** Less certain

**Metareview:**

Motivated by the often overly conservative characteristics of distributionally robust MDPs, this paper employs (nested) Bayesian posterior distributions to model the uncertainty over MDP parameters. The programming solution is similar to belief state approximation methods for POMDPs. The experiments (after revision) seem to demonstrate the advantages of this approach. The reviewers believe the paper could be improved with better theoretical analyses and/or more compelling experiments (higher dimensional tasks in particular). The paper lacks a strong advocate among the reviewers, but their aggregate sentiment is that it is worth accepting to the conference unless there are other more deserving works that it would displace.

**Award:**

No

---

### Decision · Program_Chairs · 2022-09-14

Accept